# Metage2Metabo, microbiota-scale metabolic complementarity for the identification of key species

Arnaud Belcour[1†], Clémence Frioux[1,2,3,4†*], Méziane Aite[1], Anthony Bretaudeau[1,5,6], Falk Hildebrand[3,4], Anne Siegel[1]

[1]Univ Rennes, Inria, CNRS, IRISA, Rennes, France; [2]Inria Bordeaux Sud-Ouest, Talence, France; [3]Gut Microbes and Heath, Quadram Institute, Norwich, United Kingdom; [4]Digital Biology, Earlham Institute, Norwich, United Kingdom; [5]Inria, UMR IGEPP, BioInformatics Platform for Agroecosystems Arthropods (BIPAA), Rennes, France; [6]Inria, IRISA, GenOuest Core Facility, Rennes, France

**Abstract** To capture the functional diversity of microbiota, one must identify metabolic functions and species of interest within hundreds or thousands of microorganisms. We present Metage2Metabo (M2M) a resource that meets the need for de novo functional screening of genome-scale metabolic networks (GSMNs) at the scale of a metagenome, and the identification of critical species with respect to metabolic cooperation. M2M comprises a flexible pipeline for the characterisation of individual metabolisms and collective metabolic complementarity. In addition, M2M identifies key species, that are meaningful members of the community for functions of interest. We demonstrate that M2M is applicable to collections of genomes as well as metagenome-assembled genomes, permits an efficient GSMN reconstruction with Pathway Tools, and assesses the cooperation potential between species. M2M identifies key organisms by reducing the complexity of a large-scale microbiota into minimal communities with equivalent properties, suitable for further analyses.

**\*For correspondence:**
clemence.frioux@inria.fr

[†]These authors contributed equally to this work

**Competing interests:** The authors declare that no competing interests exist.

## Introduction

Understanding the interactions between organisms within microbiomes is crucial for ecological (*Tara Oceans coordinators et al., 2015*) and health (*Integrative HMP (iHMP) Research Network Consortium, 2014*) applications. Improvements in metagenomics, and in particular the development of methods to assemble individual genomes from metagenomes, have given rise to unprecedented amounts of data which can be used to elucidate the functioning of microbiomes. Hundreds or thousands of genomes can now be reconstructed from various environments (*Pasolli et al., 2019*; *Forster et al., 2019*; *Zou et al., 2019*; *Stewart et al., 2018*; *Almeida et al., 2020*), either with the help of reference genomes or through metagenome-assembled genomes (MAGs), paving the way for numerous downstream analyses. Some major interactions between species occur at the metabolic level. This is the case for negative interactions such as exploitative competition (e.g. for nutrient resources), or for positive interactions such as cross-feeding or syntrophy (*Coyte and Rakoff-Nahoum, 2019*) that we will refer to with the generic term of cooperation. In order to unravel such interactions between species, it is necessary to go beyond functional annotation of individual genomes and connect metagenomic data to metabolic modelling. The main challenges impeding mathematical and computational analysis and simulation of metabolism in microbiomes are the scale of metagenomic datasets and the incompleteness of their data.

Genome-scale metabolic networks (GSMNs) integrate all the expected metabolic reactions of an organism. *Thiele and Palsson, 2010* defined a precise protocol for their reconstruction, associating

**eLife digest** All the microbes that live in a specific environment, for example an organ, are collectively called the microbiota. In humans, the microbiota of the gut has been extensively studied by sequencing the DNA of the different microbes to identify them and determine the roles they play in health and disease. The DNA sequences of all the members of the microbiota is called the metagenome.

The chemical reactions that the gut microbiota perform to produce energy and make the biomolecules they need to survive are collectively referred to as the metabolism of these microbes. Studying the metabolism of the gut microbiota can help researchers understand the roles of the different microbes. However, the large variety of species in the gut microbiota and gaps in the information about them render these studies difficult, despite technology improving quickly.

To tackle this issue, Belcour, Frioux et al developed a new piece of software called Metage2Metabo (M2M) that simulates the metabolism of the gut microbiota and describes the metabolic relationships between the different microbes. Metage2Metabo analyses the roles of the metabolic genes of a large number of microbe species to establish how they complement each other metabolically. Then, it can calculate the minimum number of species needed to perform a metabolic role of interest within that microbiota, and which key species are associated with that role.

To test the new software, Belcour, Frioux et al. used Metage2Metabo to analyse genomes from the human gut microbiota and from the cow rumen (one of the cow's stomachs). They showed that even if the metagenome was incomplete, the software was able to make stable predictions of key species involved in metabolic complementarity. Additionally, they also illustrated how the method can be used to study the gut microbiota of individuals.

This work presents a new method for determining the metabolic relationships between species within a microbiota. The software is highly flexible and could be adapted to identify key members within different communities. In the context of the gut microbiota, the predictions of Metage2Metabo could shed lights on the interactions between the host and the microbes and contribute to a better understanding of microbe environments.

the use of automatic methods and thorough curation based on expertise, literature, and mathematical analyses. There now exists a variety of GSMN reconstruction implementations: all-in-one platforms such as Pathway Tools (*Karp et al., 2016*), CarveMe (*Machado et al., 2018*) or KBase that provides narratives from metagenomic datasets analysis up to GSMN reconstruction with Model-SEED (*Henry et al., 2010*; *Seaver et al., 2020*). In addition, a variety of toolboxes (*Aite et al., 2018*; *Wang et al., 2018*; *Schellenberger et al., 2011*), or individual tools perform targeted refinements and analyses on GSMNs (*Prigent et al., 2017*; *Thiele et al., 2014*; *Vitkin and Shlomi, 2012*). Reconstructed GSMNs are a resource to analyse the metabolic complementarity between species, which can be seen as a representation of the putative cooperation within communities (*Opatovsky et al., 2018*). SMETANA (*Zelezniak et al., 2015*) estimates the cooperation potential and simulates flux exchanges within communities. MiSCoTo (*Frioux et al., 2018*) computes the metabolic potential of interacting species and performs community reduction. NetCooperate (*Levy et al., 2015*) predicts the metabolic complementarity between species.

In addition, a variety of toolboxes have been proposed to study communities of organisms using GSMNs (*Kumar et al., 2019*; *Sen and Orešič, 2019*), most of them relying on constraint-based modelling (*Chan et al., 2017*; *Zomorrodi and Maranas, 2012*; *Khandelwal et al., 2013*). However, these tools can only be applied to communities with few members, as the computational cost scales exponentially with the number of included members (*Kumar et al., 2019*). Only recently has the computational bottleneck started to be addressed (*Diener et al., 2020*). In addition, current methods require GSMNs of high quality in order to produce accurate mathematical predictions and quantitative simulations. Reaching this level of quality entails manual modifications to the models using human expertise, which is not feasible at a large scale in metagenomics. Automatic reconstruction of GSMNs scales to metagenomic datasets, but it comes with the cost of possible missing reactions and inaccurate stoichiometry that impede the use of constraint-based modelling (*Bernstein et al., 2019*). Therefore, development of tools tailored to the analysis of large communities is needed.

Here, we describe Metage2Metabo (M2M), a software system for the characterisation of metabolic complementarity starting from annotated individual genomes. M2M capitalises on the parallel reconstruction of GSMNs and a relevant metabolic modelling formalism to scale to large microbiotas. It comprises a pipeline for the individual and collective analysis of GSMNs and the identification of communities and key species ensuring the producibility of metabolic compounds of interest. M2M automates the systematic reconstruction of GSMNs using Pathway Tools or relies on GSMNs provided by the user. The software system uses the algorithm of network expansion (*Ebenhöh et al., 2004*) to capture the set of producible metabolites in a GSMN. This choice answers the needs for stoichiometry inaccuracy handling, and the robustness of the algorithm was demonstrated by the stability of the set of reachable metabolites despite missing reactions (*Handorf et al., 2005*; *Kruse and Ebenhöh, 2008*). Consequently, M2M scales metabolic modelling to metagenomics and large collections of (metagenome-assembled) genomes.

We applied M2M on a collection of 1520 draft bacterial reference genomes from the gut microbiota (*Zou et al., 2019*) in order to illustrate the range of analyses the tool can produce. This demonstrates that M2M efficiently reconstructs metabolic networks for all genomes, identifies potential metabolites produced by cooperating bacteria, and suggests minimal communities and key species associated to their production. We then compared metabolic network reconstruction applied to the gut reference genomes to the results obtained with a collection of 913 cow rumen MAGs (*Stewart et al., 2018*). In addition, we tested the robustness of metabolic prediction with respect to genome incompleteness by degrading the rumen MAGs. The comparison of outputs from the pipeline indicates stability of the results with moderately degraded genomes, and the overall suitability of M2M to MAGs. Finally, we demonstrated the applicability of M2M in practice to metagenomic data of individuals. To that purpose, we reconstructed communities for 170 samples of healthy and diabetic individuals (*Forslund et al., 2015*; *Diener et al., 2020*). We show how M2M can help connect sequence analyses to metabolic screening in metagenomic datasets.

## Results

### M2M pipeline and key species

M2M is a flexible software solution that performs automatic GSMN reconstruction and systematic screening of metabolic capabilities for up to thousands of species for which an annotated genome is available. The tool computes both the individual and collective metabolic capabilities to estimate the complementarity between the metabolisms of the species. Then based on a determined metabolic objective which can be ensuring the producibility of metabolites that need cooperation, that we call cooperation potential, M2M performs a community reduction step that aims at identifying a minimal community fulfilling the metabolic objective, as well as the set of associated key species.

M2M's main pipeline (*Figure 1a*) consists in five main steps that can be performed sequentially or independently: (i) reconstruction of metabolic networks for all annotated genomes, (ii) computation of individual and (iii) collective metabolic capabilities, (iv) calculation of the cooperation potential, and (v) identification of minimal communities and key species for a targeted set of compounds.

Sets of producible metabolites for individual or communities of species are computed using the network expansion algorithm (*Ebenhöh et al., 2004*) that is implemented in Answer Set Programming in dependencies of M2M. Network expansion enables the calculation of the *scope* of one or several metabolic networks in given nutritional conditions, described as seed compounds. The scope therefore represents the metabolic potential or reachable metabolites in these conditions (see Materials and methods). M2M calculates individual scopes for all metabolic networks, and the community scope comprising all reachable metabolites for the interacting species. Network expansion is also used in the community reduction optimisation implemented in MiSCoTo (*Frioux et al., 2018*), the dependency of M2M, as reduced communities are expected to produce the metabolites of interest.

The inputs to the whole workflow are a set of annotated genomes, a list of nutrients representing a growth medium, and optionally a list of targeted compounds to be produced by selected communities that will bypass the default objective of ensuring the producibility of the cooperation potential. Users can use the annotation pipeline of their choice prior running M2M. The whole pipeline is called with the command `m2m workflow` but each step can also be run individually as described in *Table 1*.

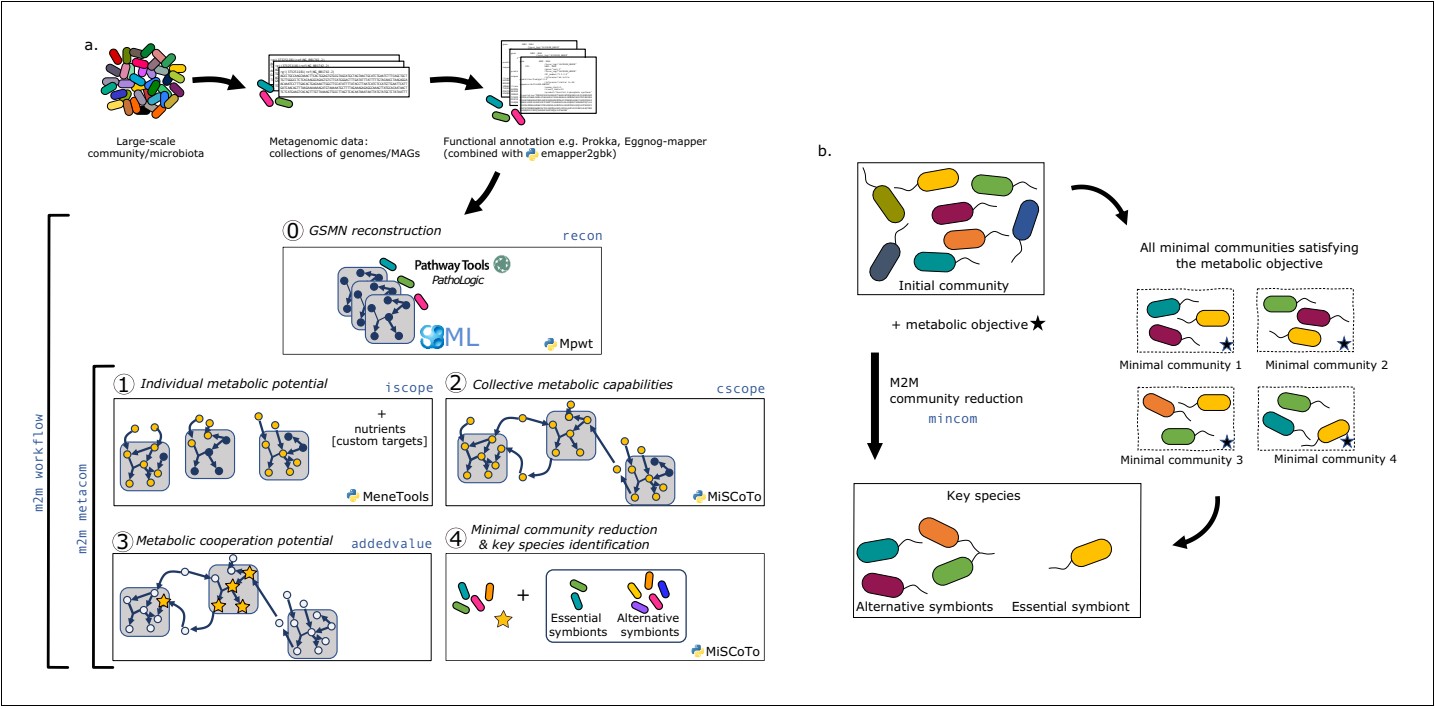

**Figure 1.** Overview of the Metage2Metabo (M2M) pipeline. (**a**) Main steps of the M2M pipeline and associated tools. The software's main pipeline (m2m workflow) takes as inputs a collection of annotated genomes that can be reference genomes or metagenomic-assembled genomes. The first step of M2M consists in reconstructing metabolic networks with Pathway Tools (step 0). This first step can be bypassed and genome-scale metabolic networks (GSMNs) can be directly loaded in M2M. The resulting metabolic networks are analysed to identify individual (step 1) and collective (step 2) metabolic capabilities. The added-value of cooperation is calculated (step 3) and used as a metabolic objective to compute a minimal community and key species (step 4). Optionally, one can customise the metabolic targets for community reduction. The pipeline without GSMN reconstruction can be called with m2m metacom, and each step can also be called independently (m2m iscope, m2m cscope, m2m addedvalue, m2m mincom). (**b**) Description of key species. Community reduction performed at step 4 can lead to multiple equivalent communities. M2M provides one minimal community and efficiently computes the full set of species that occur in all minimal communities, without the need for a full enumeration, thanks to projection modes. It is possible to distinguish the species occurring in every minimal community (essential symbionts), from those occurring in some (alternative symbionts). Altogether, these two groups form the key species.

A main characteristic of M2M is to provide at the end of the pipeline a set of *key species* associated to a metabolic function together with one minimal community predicted to satisfy this function. We define as key species organisms whose GSMNs are selected in at least one of the minimal communities predicted to fulfill the metabolic objective. Among key species, we distinguish those that

**Table 1.** List and description of Metage2Metabo (M2M) commands.

| Command | Action |
| --- | --- |
| `m2m workflow` | Runs the whole m2m workflow |
| `m2m metacom` | Runs the workflow with already-reconstructed metabolic networks |
| `m2m recon` | Reconstructs metabolic networks using Pathway Tools |
| `m2m iscope` | Computes scopes for individual metabolic networks |
| `m2m cscope` | Computes the community scope |
| `m2m addedvalue` | Computes the cooperation potential |
| `m2m mincom` | Selects a minimal community and computes key species |
| `m2m seeds` | Creates a SBML file for nutrients |
| `m2m test` | Runs m2m workflow on a sample dataset |
| `m2m_analysis` | Runs additional analyses on community selection |

occur in every minimal community, suggesting that they possess key functions associated to the objective, from those that occur only in some communities. We call the former *essential symbionts*, and the latter *alternative symbionts*. These terms were inspired by the terminology used in flux variability analysis (*Orth et al., 2010*) for the description of reactions in all optimal flux distributions. If interested, one can compute the enumeration of all minimal communities with *m2m_analysis*, which will provide the total number of minimal communities as well as the composition of each. *Figure 1b* illustrates these concepts with an initial community formed of eight species. There are four minimal communities satisfying the metabolic objective. Each includes three species, and in particular, the yellow one is systematically a member. Therefore, the yellow species is an essential symbiont whereas the four other species involved in minimal communities constitute the set of alternative symbiont. As key species represent the diversity associated to all minimal communities, it is likely that their number is greater than the size of a minimal community, as this is the case in *Figure 1b*.

## M2M connects metagenomics to metabolism with GSMN reconstruction, metabolic complementarity screening and community reduction

In order to illustrate its applicability to real data, M2M was applied to a collection of 1520 bacterial high-quality draft reference genomes from the gut microbiota presented in *Zou et al., 2019*. The genomes were derived from cultured bacteria, isolated from faecal samples covering typical gut phyla (*Costea et al., 2018*): 796 Firmicutes, 447 Bacteroidetes, 235 Actinobacteria, 36 Proteobacteria, and 6 Fusobacteria. The dereplicated genomes represent 338 species. The genomes were already annotated and could therefore directly enter M2M pipeline. The full workflow (from GSMN reconstruction to key species computation) took 155 min on a cluster with 72 CPUs and 144 Gb of memory. We illustrate in the next paragraphs the scalability of M2M and the range of analyses it proposes by applying the pipeline to this collection of genomes.

### GSMN reconstruction

GSMNs were automatically reconstructed for the 1520 isolate-based genomes using their published annotation. A total of 3932 unique reactions and 4001 metabolites were included in the reconstructed GSMNs (*Table 2*). The reconstructed gut metabolic networks contained on average 1144 (±255) reactions and 1366 (±262) metabolites per genome. Of the reactions, 74.6% were associated to genes, the remaining being spontaneous reactions or reactions added by the PathoLogic algorithm (they can be removed in M2M using the *–noorphan* option).

**Table 2.** Results of the genome-scale metabolic network (GSMN) reconstruction step and metabolic potential analysis for the three datasets presented in the article (Avg = Average, '±' precedes standard deviation).

|  | Gut dataset | Rumen dataset | Diabetes dataset |
| --- | --- | --- | --- |
| Initial data | Draft reference genomes | MAGs | MAGs |
| Number of genomes | 1520 | 913 | 778 |
| *GSMN reconstruction* |  |  |  |
| All reactions | 3932 | 4418 | 5554 |
| All metabolites | 4001 | 4466 | 5386 |
| Avg reactions per GSMN | 1144 (±255) | 1155 (±199) | 1640 (±368) |
| Avg metabolites per GSMN | 1366 (±262) | 1422 (±212) | 1925 (±361) |
| Avg genes per mn | 596 (±150) | 543 (±107) | 1658 (±469) |
| % reactions associated to genes | 74.6 (±2.17) | 73.8 (±2.61) | 79.57 (±1.60) |
| Avg pathways per mn | 163 (±49) | 146 (±32) | 220 (±58) |
| *Metabolic potential* |  |  |  |
| Number of seeds | 93 | 26 | 175 |
| Avg scope per mn | 286 (±70) | 101 (±44) | 508 (±83) |
| Union of individual scopes | 828 | 368 | 1326 |

The metabolic potential, or *scope*, was computed for each individual GSMN (*Table 2*). Nutrients in this experiment were components of a classical diet (see Materials and methods). The union of all individual scopes is of size 828 (21% of all compounds included in the GSMNs), indicating a small part of the metabolism reachable in the chosen nutritional environment (*Supplementary file 1* - Table 1). *Appendix 1—figure 1h,i* displays the distributions of the scopes. Across all GSMNs, individual scopes are overall stable in size. The core set of producible metabolites is small and a variety of metabolites are only reachable by a small number of organisms (*Appendix 1—figure 1i*). The overall small size of metabolic potentials can be explained by the restricted amount of seeds used for computation. Among metabolites that are reachable by all or almost all metabolic networks, the primary metabolism is highly represented, as expected, with metabolites derived from common sugars (glucose, fructose), pyruvate, 2-oxoglutaric acid, amino acids... On the other hand but not surprisingly, metabolites that predicted to be reached by a limited number of individual producers include compounds from secondary metabolism: (fatty) acids (e.g. oxalate, maleate, allantoate, hydroxybutanoate, methylthiopropionate), derivatives of amino acids, amines (spermidine derivatives).

## Cooperation potential

Metabolic cooperation enables the activation of more reactions in GSMNs than what can be expected when networks are considered in isolation. By taking into account the complementarity between GSMNs in each dataset, it is possible to capture the putative benefit of metabolic cooperation on the diversity of producible metabolites. Running *m2m cscope* predicted 156 new metabolites as producible by the gut collection of GSMNs if cooperation is allowed.

We analysed the composition of the 156 newly producible metabolites for the gut dataset using the ontology provided for metabolic compounds in the MetaCyc database. 80.1% of them could be grouped into six categories: amino acids and derivatives (5 metabolites), aromatic compounds (11), carboxy acids (14), coenzyme A (CoA) derivatives (10), lipids (28), sugar derivatives (58). The groups were used in the subsequent analyses. The remaining 30 compounds were highly heterogeneous, we therefore restrained our subsequent analyses to subcategories of biochemically homogeneous targets.

We paid a particular attention to the predicted producibility of short-chain fatty acids (SCFAs) among genomes of the gut collection. We analysed formate, acetate, propionate, and butyrate in individual and collective metabolic potentials (*Supplementary file 1* - Table 25). A total of 543 metabolic networks are predicted to be able to produce all four molecules in a cooperative context, 74% of them belonging the Firmicutes, as expected. Surprisingly, predicted individual producers of the four SCFAs (n = 128) are mostly Bacteroidetes (70%) suggesting the dependency of Firmicutes to interactions in order to permit the producibility of SCFAs in this experimental setting. The same observations are made when focusing on butyrate alone, that has the particularity of belonging to the seeds. As Bacteroidetes are not the main butyrate producers in the gut, the predictions of such producibility is likely an artefact relying on alternative pathways, and further emphasises the fact that owning the genetic material for a function does not entail its expression.

## Key species associated to groups of metabolites

M2M proposes by default one community composition for an objective defined by enabling the producibility of metabolic end-products. Given the functional redundancy of gut bacteria (*Moya and Ferrer, 2016*), there could be thousands of bacterial composition combinations, and it is computationally costly to enumerate them. To circumvent this restriction, M2M identifies *key species* without the need for all possible combinations of species to be enumerated, consequentially reducing computational time. Key species include all species occurring in at least one minimal community for the production of chosen end-products. They can be distinguished in two categories: *essential symbionts* occurring in all minimal communities, and *alternative symbionts* occurring in some minimal communities.

To explore the spectrum of possible key species, we ran M2M community reduction step (`m2m mincom` command) with the above six metabolic target groups. This allowed us to compute predicted key species for each of them (*Table 3*). The contents of key species for each of the six groups of targets as well as for the complete set of targets is displayed in *Supplementary file 1*

**Table 3.** Community reduction analysis of the target categories in the gut.

All minimal communities were enumerated, starting from the set of 1520 genome-scale metabolic networks (GSMNs). KS: key species, ES: essential symbionts, AS: alternative symbionts, Firm.: Firmicutes, Bact.: Bacteroidetes, Acti.: Actinobacteria, Prot.: Proteobacteria, Fuso.: Fusobacteria.

|  |  | Firm. | Bact. | Acti. | Prot. | Fuso. | Total |
|---|---|---|---|---|---|---|---|
| **Aminoacids and derivatives** (5 targets) 4 bact. per community 120,329 communities | KS | 142 | 52 | 0 | 27 | 6 | 227 |
|  | ES | 0 | 0 | 0 | 0 | 0 | 0 |
|  | AS | 142 | 52 | 0 | 27 | 6 | 227 |
| **Aromatic compounds** (11 targets) 5 bact. per community 950 communities | KS | 52 | 0 | 0 | 20 | 0 | 72 |
|  | ES | 2 | 0 | 0 | 1 | 0 | 3 |
|  | AS | 50 | 0 | 0 | 19 | 0 | 69 |
| **Carboxyacids** (14 targets) 9 bact. per community 48,412 communities | KS | 16 | 13 | 0 | 28 | 2 | 59 |
|  | ES | 2 | 0 | 0 | 2 | 0 | 4 |
|  | AS | 14 | 13 | 0 | 26 | 2 | 55 |
| **CoA derivatives** (10 targets) 5 bact. per community 95,256 communities | KS | 106 | 0 | 50 | 17 | 1 | 174 |
|  | ES | 0 | 0 | 0 | 0 | 1 | 1 |
|  | AS | 106 | 0 | 50 | 17 | 0 | 173 |
| **Lipids** (28 targets) 7 bact. per community 58,520 communities | KS | 3 | 140 | 22 | 20 | 0 | 185 |
|  | ES | 3 | 0 | 0 | 1 | 0 | 4 |
|  | AS | 0 | 140 | 22 | 19 | 0 | 181 |
| **Sugar derivatives** (58 targets) 11 bact. per community 7,860,528 communities | KS | 11 | 30 | 78 | 23 | 0 | 142 |
|  | ES | 5 | 0 | 0 | 0 | 0 | 5 |
|  | AS | 6 | 30 | 78 | 23 | 0 | 137 |

(Tables 6 to 12). To our surprise, the size of the minimal community is relatively small for each group of metabolites (between 4 and 11), compared to the initial community of 1520 GSMNs. The number of identified key species varies between 59 and 227, which might be closer to the total taxonomic diversity found in the human gut microbiome. This strong reduction compared to the initial number of 1520 GSMNs used for the analysis illustrates the existence of groups of bacteria with specific metabolic capabilities. In particular, essential symbionts are likely of high importance for the functions as they are found in each solution. More generally, compositions vary across the target categories: a high proportion of key species for the production of lipids targets are Bacteroidetes, whereas Firmicutes were more often key species for aminoacids and derivatives production. The propensity of Bacteroidetes to metabolise lipids has been proposed previously, it has for example been observed in the *Bacteroides* enterotype for functions related to lipolysis (*Vieira-Silva et al., 2016*).

## Analysis of minimal communities identifies groups of organisms with equivalent roles

To go further, we enumerated all minimal communities for each individual group of targets using *m2m_analysis*. The number of optimal solutions is large, reaching more than 7 million equivalent minimal communities for the sugar-derived metabolites (*Table 3*). Our analysis of key species indicates that the large number of optimal communities is due to combinatorial choices among a rather small number of bacteria (*Table 3*).

In order to visualise the association of GSMNs in minimal communities, we created for each target set a graph whose nodes are the key species (*Supplementary file 1* - Tables 6 to 12), and whose edges represent the association between two species if they co-occur in at least one of the enumerated communities. Graphs were very dense: 185 nodes, 6888 edges for the lipids, 142 nodes, and 6602 edges for the sugar derivatives. This density is expected given the large number of optimal communities and the comparatively small number of key species. The graphs were compressed into power graphs to capture the combinatorics of association within minimal communities. Power

graphs enable a lossless compression of re-occurring motifs within a graph: cliques, bicliques, and star patterns (*Royer et al., 2008*). The increased readability of power graphs permits pinpointing metabolic equivalency between members of the key species with respect to the target compound families.

*Figure 2* presents the compressed graphs for each set of targets. Graph nodes are the key species, coloured by their phylum. Nodes are included into power nodes that are connected by power edges, illustrating the redundant metabolic function(s) that species provide to the community when considering specific end-products. GSMNs belonging to a power node play the same role in the construction of the minimal communities. In this visualisation, essential symbionts are easily

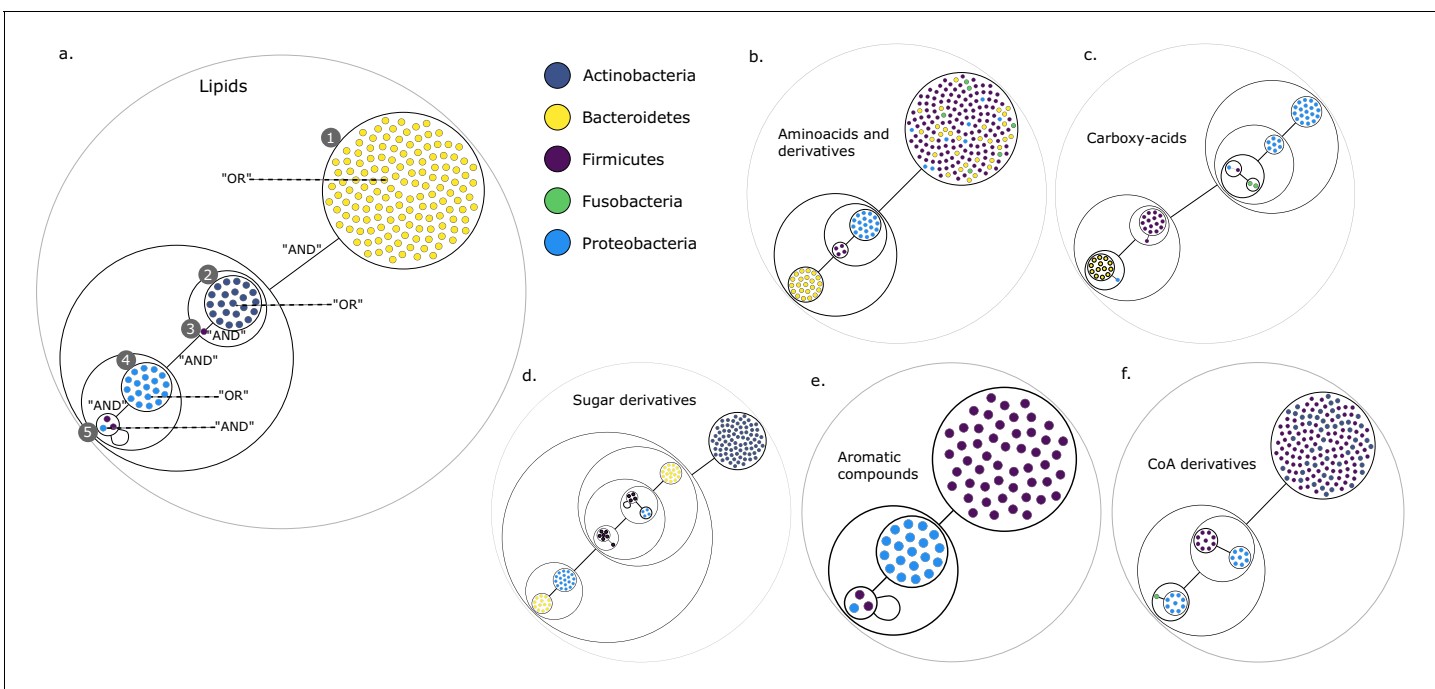

**Figure 2.** Power graph analysis of predicted microbial associations within communities for the human gut dataset. Each category of metabolites predicted as newly producible in the gut was defined as a target set for community selection among the 1520 genome-scale metabolic networks (GSMNs) from the gut microbiota reference genomes dataset. For each metabolic group, key species and the full enumeration of all minimal communities were computed. Association graphs were built to associate members that are found together in at least one minimal community among the enumeration. These graphs were compressed as power graphs to identify patterns of associations and groups of equivalence within key species. Power graphs a., b., c., d., e., f. were generated for the sets of lipids, aminoacids and derivatives, carboxy-acids, sugar derivatives, aromatic compounds, and coenzyme A derivative compounds, respectively. Node colour describes the phylum associated to the GSMN. Figure (a) has an additional description to ease readability. Edges symbolise conjunctions ('AND') and the co-occurrences of nodes in regular power nodes (as in power node 1, 2, 4) symbolise disjunctions ('OR') related to alternative symbionts. Power nodes with a loop (e.g. power node 5) indicate conjunctions. Therefore, each enumerated minimal community for lipid production is composed of the two Firmicutes and the Proteobacteria from power node 5, the Firmicutes node 3 (the four of them being the essential symbionts), and one Proteobacteria from power node 4, one Actinobacteria from power node 2 and 1 Bacteroidetes from power node 1. Members from an inner power node are interchangeable with respect to the metabolic objective. A version of the figures with species identification is available in *Figure 2—figure supplement 1*, *Figure 2—figure supplement 2*, *Figure 2—figure supplement 3*, *Figure 2—figure supplement 4*, *Figure 2—figure supplement 5*, *Figure 2—figure supplement 6* (see *Supplementary file 1* - Table 4 for a mapping between identifiers and taxonomy). Power graphs can be generated with m2m_analysis. The figures display one visual representation for each power graph although such representations are not unique. The number of power edges is minimal, which leads to nesting of (power) nodes. The online version of this article includes the following figure supplement(s) for figure 2:

**Figure supplement 1.** Sugars derivatives power graph.

**Figure supplement 2.** Lipids derivatives power graph.

**Figure supplement 3.** Amino acids and derivatives power graph.

**Figure supplement 4.** Aromatic compounds power graph.

**Figure supplement 5.** Carboxy-acids compounds power graph.

**Figure supplement 6.** Coenzyme A derivatives power graph.

identifiable, either into power nodes with loops (*Figure 2a,e*) or as individual nodes connected to power nodes (*Figure 2a,c,d,f*).

We observe that power nodes often contain GSMNs from the same phylum, indicating that phylogenetic groups encode redundant functions. *Figure 3a* has additional comments to guide the reader into analysing the community composition on one example. Each minimal community suitable for the production of the targeted lipids is composed of one Bacteroidetes from power node (PN) 1, one Actinobacteria from PN 2, the Firmicutes member 3, one Proteobacteria from PN 4 and finally the two Firmicutes and the Proteobacteria from PN 5. For all the target groups of this study, the

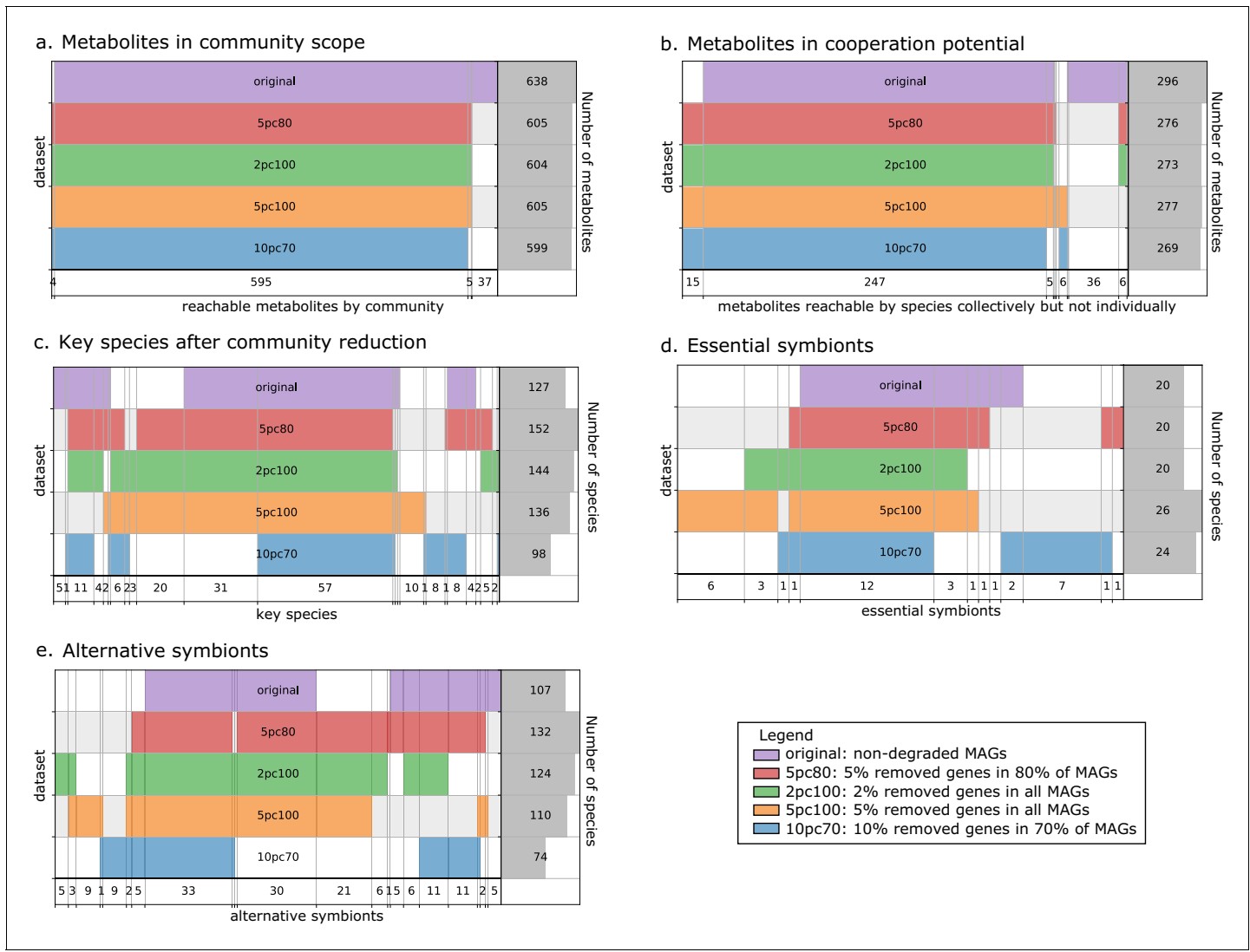

**Figure 3.** Robustness analysis of Metage2Metabo (M2M) results on datasets of altered metagenome-assembled genomes (MAGs). A proportion of genes were randomly removed from all or a random subset of the 913 rumen MAGs: 2% from all genomes (*2pc100*), 5% from 80% of the genomes (*5pc80*), 5% from all genomes (*5pc100*) and 10% from 70% of the genomes (*10pc70*). M2M pipeline was ran on these four datasets and comparison was made with respect to the initial non-altered dataset of MAGs (*original*). Subfigures *a* to *e* each represent one piece of information computed by M2M and compared between the five experiments. (**a**) Set of producible compounds by all metabolic networks in a cooperative system (community scope); supervenn representation. Each dataset of metabolic networks obtained from the original or degraded genomes is represented horizontally, with a unique colour. The right panel of the supervenn diagram indicates the number of metabolites in the community scope of the corresponding dataset. Vertical overlaps between sets represent intersections (e.g groups of metabolites retrieved in several datasets) whose size is indicated on the X axis. For example, there is a set of 37 metabolites that are producible in the original dataset only, and a set of 5 metabolites predicted as producible in all datasets but the one where 70% of genomes were 10%-degraded. A full superimposition of all the coloured bars would indicate a complete stability of the community scope between datasets. (**b**) Comparison of the cooperation potential between the five experiments. (**c**) Comparison of key species that gather essential symbionts (**d**) and alternative symbionts (**e**).

large enumerations can be summarised with a boolean formula derived from the graph compressions. For instance, for the lipids of *Figure 3a*, the community composition as described above is the following:

$$(\vee PN1) \wedge (\vee PN2) \wedge (PN3) \wedge (\vee PN4) \wedge (\wedge PN5).$$

We further investigated the essential symbionts associated to carbohydrate-derived metabolites in our study: *Paenibacillus polymyxa*, *Lactobacillus lactis*, *Bacillus licheniformis*, *Lactobacillus plantarum*, and *Dorea longicatena*. Interestingly, out of these five species, the first four have already been studied in the context of probiotics, for animals or humans (*Cutting, 2011*; *Monteagudo-Mera et al., 2012*). In particular, the study of *P. polymyxa* CAZymes demonstrated its ability to assist in digesting complex carbohydrates (*Soni et al., 2020*). *L. plantarum* is also known for its role in carbohydrate acquisition (*de Vries et al., 2006*; *Marco et al., 2010*). The present analysis illustrates that within the full genome collection, these species are likely to exhibit functions related to carbohydrate synthesis and degradation that are not found in other species. *Bacillus licheniformis* is also an essential symbiont for the lipid metabolites. Among essential symbionts for other groups of metabolites, *Burkholderiales bacterium* (Proteobacteria) and *Hungatella hathewayi* (Firmicutes) have the particularity of occurring in predictions for the lipids, carboxy acids, and aromatic metabolites. This suggest a metabolism for these two species that differs from the other species, with non-redundant contributions to some metabolites of these categories. While *Hungatella hathewayi* is a relatively frequent gut commensal, little is known about this species (*Manzoor et al., 2017*). The *Burkholderiales* order is also poorly known, but its ability to degrade a variety of aromatic compounds has been established (*Pérez-Pantoja et al., 2012*). Finally, the only essential symbiont predicted for the coA-related metabolites is *Fusobacterium varium*, a butyrate producer known for its ability to ferment both sugars and amino acids (*Potrykus et al., 2008*).

Altogether, computation of key species coupled to the visualisation of community compositions enables a better understanding of the associations of organisms into the minimal communities. In this genome collection, groups of equivalent GSMNs allow us to identify genomes that are providing specialised functions to the community, enabling metabolic pathways leading to specific end-products.

## M2M is suited to the metabolic analysis of MAGs

### Comparison of M2M applications to MAGs and draft reference genomes

In order to compare the effect of genome quality on M2M predictions, we performed analyses on a collection of 913 MAGs binned from cow rumen metagenomes (*Stewart et al., 2018*). These MAGs were predicted to be >80% complete and <10% contaminated. The complete M2M workflow ran in 81 min on a cluster with 72 CPUs and 144 Gb of memory.

Results of the GSMN reconstruction are presented in *Table 2*. GSMNs of the cow rumen MAGs dataset consisted in average of 1155 (±199) reactions and 1422 (±212) metabolites. 73.8% of the reactions could be associated to genes. We compared these numbers with those obtained for the collection of draft reference genomes from the human gut microbiota of the previous subsection. Appendix 1 displays the distributions of the numbers of reactions, pathways, metabolites and genes for both datasets. Altogether, these distributions are very similar for both datasets although the initial number of genes in the whole genomes varies a lot (Appendix 1 g), a difference that is expected between MAGs and reference genomes. Interestingly, the average number of reactions per GSMN is slightly higher for the MAGs of the rumen than for the reference genomes of the gut. This could be explained by the higher phylogenetic diversity observed in MAGs compared to culturable bacteria, or a higher potential for contaminated genomes, or a difference in average genome size. However, the smallest GSMN size is observed in the rumen (340 reactions vs 617 for the smallest GSMN of the gut dataset). The similarity in the characteristics displayed by both datasets suggests a level of quality of the rumen MAGs close to the one of the gut reference genomes regarding the genes associated to metabolism. This is consistent with the high-quality scores of the MAGs described in the original publication: the 913 MAGs exhibited a CheckM (*Parks et al., 2015*) completeness score between 80% and 100% (average: 90.61%, standard deviation: 5.26%, median: 91.03%) (*Stewart et al., 2018*).

M2M modelling analyses were run on the reconstructed GSMNs. Appendix 1 (*Figure 1*; *Appendix 1—figure 1*) displays the distributions of the individual scopes for each GSMN. We identified a cooperation potential of 296 metabolic end-products only reachable through the community. The minimal community consisted of 44 GSMNs, sufficient to produce all metabolites reachable through cooperation in the initial community composition. These could be described through 127 key species, consisting of 20 essential symbionts and 107 alternative ones. This indicates that each equivalent minimal community for these compounds would consist in the same 20 GSMNs, associated to 24 others selected within the 107 alternative species, thereby reaching a total of 44 GSMNs. Results are displayed in *Supplementary file 1* (Table 4), together with those of an equivalent analysis (default settings of M2M with cooperation potential as targets) for the gut reference genomes.

## M2M robustly identifies key species, even with degraded genomes

A recurring concern in metagenomics is the completeness of reconstructed MAGs due to the possible loss of functions during the genome assembly process (*Parks et al., 2015*). Misidentified genes can impede GSMN reconstruction and consequently the contents of the scopes and cooperation potential. To assess the impact of MAG completeness, we altered the rumen MAGs dataset by randomly removing genes. We created four altered datasets by removing: (i) 2% of genes in all genomes, (ii) 5% of genes in 80% of the genomes, (iii) 5% of genes in all genomes and (iv) 10% of genes in 70% of the genomes. We analysed these degraded datasets with the same M2M workflow, using a community selection with the metabolic cooperation potential as a community objective.

The metabolic cooperation potential, the global set of reachable metabolites in the community and the key species (essential and alternative symbionts) were computed and compared between the four altered datasets and the original one. Results are depicted in *Figure 3*. The global set of producible metabolites in the community and the contents of the metabolic cooperation potential remain stable between datasets. In both we observe a single subset of 36 metabolites that is only reachable by the original dataset. It consists in a variety of metabolites mostly from secondary metabolism. Discrepancies appear between datasets when studying key species with respect to the original dataset, with an overall stability of the datasets to 2% degradation and to 5% degradation in 80% of genomes. The main discrepancies are observed for alternative symbionts with an additional small set of symbionts that are selected in altered datasets but not in the original one. For the most degraded genomes (10% degradation in 70% of MAGs), key species composition is altered compared to original genomes: a set of 31 key species is no longer identified. However, producibility analyses and community selections performed by M2M are stable to small genome degradations of up to 2% of random gene loss in all genomes or 5% in 80% of the genomes. Altogether, *Figure 3* illustrates relative stability of the information computed by M2M to missing genes. The criteria typically used for MAG quality (>80% completeness, <10% degradation) are likely sufficient to get a coarse-grained, yet valuable first picture of the metabolism. This robustness in our algorithms could be explained by the fact that (i) missing genes in degraded MAGs may not be related to metabolism, (ii) by the reported stability of the network expansion algorithm to missing reactions (*Handorf et al., 2005*), and (iii) by the fact that multiple genes can be associated to the same metabolic reaction (redundancy in pathway representation). Additional analyses (Appendix 1) enable the refutation of the first hypothesis as the average gene loss in metabolic networks is similar to the genomic loss. Yet, the percentage of reactions associated to genes is similar in every experiments, which goes in the direction of the redundancy loss hypothesis. Likewise, we observed that the loss in reactions for degraded genomes is lower than the loss of genes.

## Application of M2M to human shotgun metagenomic data from diabetic and healthy individuals

### Protocols and cohort effect

In order to illustrate the applicability of M2M to metagenomic samples and cohorts of individuals, we reused the work presented in *Diener et al., 2020* and analysed the gut metagenomes of 170 individuals from a Danish (MHD) cohort and a Swedish (SWE) cohort (*Forslund et al., 2015*) in the context of Type-1 (T1D) and Type-2 (T2D) diabetes. Based on species-level dereplicated MAGs, metagenomic species (MGS), we built GSMNs and bacterial communities for each individual. We relied only on the available metagenomic data to perform analyses, and used qualitative information

(presence/absence of MGS or species in the sample) to build the communities as M2M works with qualitative information. Two experiments were performed: firstly, M2M was run on each sample using communities of newly reconstructed GSMN from MGS, and secondly using communities consisting of curated GSMNs of the AGORA resources (*Magnúsdóttir et al., 2017*) mapped to OTUs at the species level as described in *Diener et al., 2020*. A total of 778 MGS were retrieved from the dataset and used to build GSMNs (*Table 2*), whereas when using the mapping of OTUs to existing curated GSMNs, only 289 GSMNs were used. The distribution of phyla in the two cases is illustrated in *Appendix 2—figure 2*. We first focus on the results obtained with the MGS-based protocol.

In average, communities were composed of 108 (±29) GSMNs. The median community size was 111 (*Supplementary file 1* - Table 20). Diversity and richness analyses are available in Appendix 2 The effect of the cohort (MHD, SWE) was strong in the analyses performed with MGS, impacting community sizes, size and composition (in families of metabolites) of the set of metabolites producible by the community, as well as the cooperation potential. Results are depicted in *Appendix 2—figure 3*. A classification experiment using the composition of the community scope or the composition of the cooperation potential can efficiently determine the cohort of the samples (*Appendix 2—figure 3* panels c and g). Similar differences between cohorts were also observed in *Diener et al., 2020* and in *Forslund et al., 2015* based on functional or taxonomic annotations, likely driven by the different sampling protocols used in the two datasets (*Forslund et al., 2015*). This indicates that the commonly observed cohort effect in metagenomics is also reflected at the metabolic modelling scale, which could be explained by the observed GSMN redundancies shared within phyla.

## Impact of the disease status

We studied the impact of the disease status on the community metabolism for the 115 samples of the MHD cohort. The community diversity varied between disease statuses, with a significantly higher number of MGS observed in T1D individuals forming the initial communities (anova $F_{(2,112)}$ = 8.346, $p<0.01$, eta-squared = 0.13, Tukey HSD test $p<0.01$ vs control). We observe that the distribution of the community sizes is broader for control individuals. The higher diversity for diseased individuals is reflected at the metabolic level through the putative producibility of a wider set of metabolites for T1D (anova $F_{(2,112)}$ = 6.606, $p<0.01$, eta-squared = 0.11, Tukey HSD $p<0.01$ vs control) and to a lesser extent for T2D communities (Tukey HSD $p=0.05$ vs control). The putative producibility of some families of metabolites (alcohols, esters, carbohydrates, amino acids, acids) in the community scopes also differed between metabolic communities derived from diseased and healthy individuals (anova $p<0.05$), whereas other metabolic families like lipids remained stable between cohorts. This can be at least partly explained by the number of metabolites matching these categories according to the Metacyc database (e.g. 191 metabolites tagged as 'All-carbohydrates' in average in community scopes, and only 10 tagged as 'Lipids' as the remaining of them are scattered in other categories). No clear difference appears between the three statuses (*Figure 4e*) in terms of community scope composition. Regarding the cooperation potential, two groups tend to appear, separated due to diverse secondary metabolites, but they are not driven by the disease status of the individual (*Figure 4f*). A classification experiment on the composition of the community scope can, to some extent (AUC = 0.75 ± 0.15), decipher between healthy or diabetes statuses (*Figure 4d*) but classification between T1D and T2D was not achievable (*Appendix 2—figure 4*). Although metagenomic data would more precisely perform such a separation, it is informative to observe that despite metabolic redundancy in the gut microbiota, there are differences at the metabolic modelling level. Qualitative differences are noticeable between healthy and diabetic individuals: it is possible to distinguish them to some extent using the set of metabolites predicted to be producible by the microorganisms found in their faeces.

We then computed for each sample the key species (essential and alternative symbionts) associated to the cooperation potential. The ratio of key species (KS), essential symbionts (ES) and alternative symbionts (AS) with respect to the initial community size did not vary altogether between statuses. The exception was the ratio of AS (and of KS, which include AS) when comparing diabetes individuals and controls, differences that were not significant when distinguishing the two types of diabetes. Comparing the phylum-level taxonomy of these putative key species, in the initial communities, we noted that the occurrence of Firmicutes was broader compared to other phyla (Bacteroidota, Proteobacteria, Actinobacteria). Firmicutes are known to be phylogenetically diverse

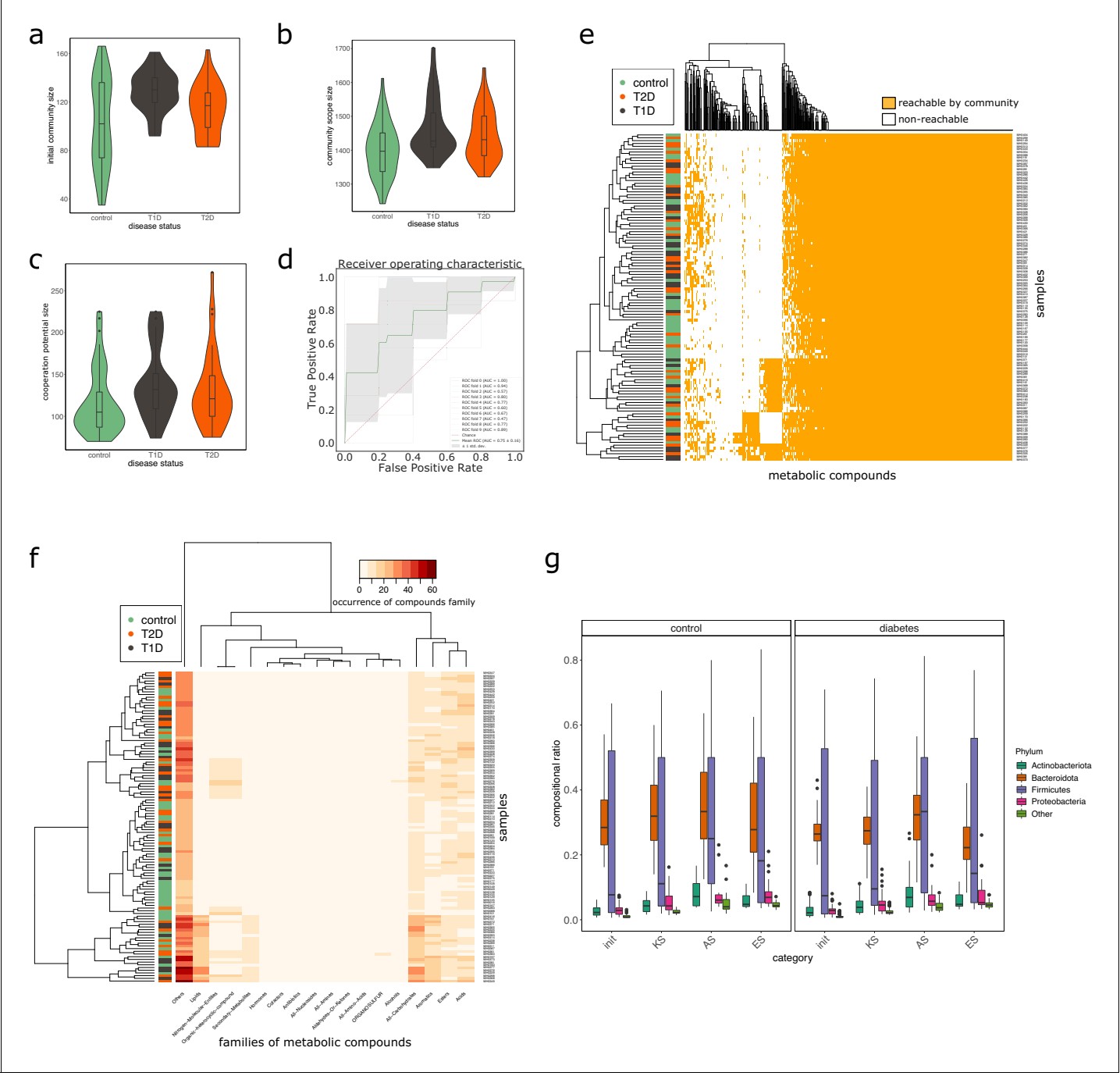

**Figure 4.** Effect of the disease status on the metabolism of communities in MHD samples. M2M was run on collections of genome-scale metabolic networks (GSMNs) associated to metagenome-assembled genomes (MAGs) identified in metagenomic samples from a cohort of healthy and diabetic individuals. Panel (a) describes the distributions of community sizes for all metagenomic samples according to the disease status: T1D: Type-1 Diabetes, T2D: Type-2 Diabetes. Panels (b) and (c) show the distribution of the community scope sizes and cooperation potential sizes respectively, according to the disease status. Panel (d) is the receiver operating curve (ROC) of an SVM classification experiment aiming at predicting the disease status for the MHD cohort (control n = 49 or diabetes n = 66) based on the community scope composition. Panel (e) illustrates the community scope composition in terms of metabolites for all samples. Disease status is indicated by the colour at the left side of each row. Panel (f) illustrates the composition of the cooperation potential according to the belonging of metabolites to Metacyc families of compounds. Disease status is indicated by the colour at the left side of each row. Panel (g) describes the taxonomic distribution at the phylum level of groups of species before and after community reduction, according to the disease status. Selection of communities was performed with the objective of making producible by the reduced communities the set of metabolites in the cooperation potential. init: initial composition of communities, KS: key species, AS: alternative symbionts, ES: essential symbionts. T1D: Type-1 Diabetes, T2D: Type-2 Diabetes.

(*Costea et al., 2018*), and therefore their combined metabolism could also be more diverse. A notable change is the narrower distribution of Bacteroidota in the initial communities as well as in selected symbionts in diseased individuals compared to control (*Figure 4g*). Altogether, no clear trend was observable from metabolic modelling analyses between disease states, but we observed some difference for the taxonomic composition of minimal communities, which could be explained by the diversity discrepancies in microbiome compositions (*Forslund et al., 2015*).

## Focus on short-chain fatty acids production

Given the importance of SCFAs in human health (*Baxter et al., 2019*), we focused on the production of butyrate, propionate and acetate in communities for each sample of the dataset. A small number of MGS (N = 11) GSMNs were predicted to be able to individually produce butyrate from the nutrients. All 778 MGS were capable to ferment acetate and most of them propionate (N = 515). The putative production of butyrate in the 170 communities when allowing cooperation between GSMNs was systematic. As expected (*Rivière et al., 2016*), a majority (54.1%) of the unique MGS predicted as possible butyrate producers in communities (GSMNs comprising a reaction producing butyrate that could be activated in a community) belonged to the Firmicutes phylum. Altogether, in 62.6% of cases, the putative butyrate producers observed in the communities were Firmicutes. We compared the number of putative butyrate producers in communities from MHD samples according to the disease status of the individuals. Their number was significantly higher in the communities of T1D individuals compared to control and T2D (anova F(2,111) = 9.27, p<0.01, eta-squared = 0.14, and Tukey HSD test p<0.01 vs control and p=0.02 vs T2D) which could be explained by the higher MGS diversity observed in T1D communities compared to the others. We then analysed the difference between using GSMNs of MGS reconstructed from metagenomic data and using curated GSMNs mapped at the species level to OTUs as performed in *Diener et al., 2020*. The same increase in butyrate producers was observed when running M2M on MHD communities consisting of the mapped AGORA GSMNs (anova F(2,112) = 5.368, p<0.01, eta-squared = 0.11, and Tukey HSD test p<0.01 vs control). To conclude, similar to the analyses of *Diener et al., 2020*, we observe that the producibility of SCFAs, particularly butyrate, is highly driven by cooperation in the microbial communities of individuals and can be performed by heterogenous sets of commensal species. The MGS-driven approach and the systematic GSMN reconstruction permit taking advantage of the whole metagenomic information and capturing the metabolic complementarity in each sample.

## Discussion

M2M is a new software system for the functional analysis of metagenomic datasets at the metabolic level. M2M can be used as an all-in-one pipeline or as independent steps in order to depict an initial picture of metabolic complementarity within a community. It connects directly to metagenomics through the automation of GSMN reconstruction, and integrates in this collective analysis community reduction with respect to targeted functions. M2M was applied to a large collection of gut microbiota reference genomes, demonstrating the scalability of the methods and how it can help identifying equivalence classes among species for the producibility of metabolite families. We showed that metabolic networks reconstructed from reference genomes and MAGs display similar characteristics and that M2M modelling predictions are robust to missing genes in the original genomes. Finally, application to real metagenomic samples of individuals demonstrated that qualitative modelling of metabolism retrieves known features from metagenomics and quantitative modelling analyses. M2M provides a first order analysis with a minimal cost in terms of required data and computational effort.

The identification of cornerstone taxa in microbiota is a challenge with many applications, for instance restoring balance in dysbiotic environments. Keystone species, a concept introduced in ecology, are particularly looked for as they are key drivers of communities with respect to functions of interest (*Banerjee et al., 2018*). There is a variety of techniques to identify them (*Carlström et al., 2019*; *Floc'h et al., 2020*), and computational biology has a major role in it (*Fisher and Mehta, 2014*; *Berry and Widder, 2014*). The identification of alternative and essential symbionts by M2M is an additional solution to help identify these critical species. In particular, essential symbionts are close to the concept of keystone species as they are predicted to have a role in every minimal community associated to a function. Additionally, alternative species and the study of

their combinations in minimal communities, for example with power graphs, are also informative as they reveal equivalence groups among species.

M2M functionally analyses large collections of genomes in order to obtain metabolic insights into the metabolic complementarity between them. While the functionality of metagenomic sequences is commonly analysed at higher levels by directly computing functional profiles from reads (*Franzosa et al., 2018*; *Silva et al., 2016*; *Sharma et al., 2015*; *Petrenko et al., 2015*), the metabolic modelling oriented approach provides more in-depth predictions on reactions and pathways organisms could catalyse in given environmental conditions. M2M answers to the upscaling limitation of individual GSMN reconstruction with Pathway Tools by automating this task using the Mpwt wrapper. GSMNs in SBML format obtained from other platforms such as Kbase (*Arkin et al., 2018*), ModelSEED (*Henry et al., 2010*; *Seaver et al., 2020*), or CarveMe (*Machado et al., 2018*), can also be used as inputs to M2M for all metabolic analyses. For instance, we used highly curated models from AGORA in the application of M2M to metagenomic datasets. The above reconstruction platforms already implement solutions to facilitate the treatment of large genomic collections. There is no universal implementation for GSMN reconstruction (*Mendoza et al., 2019*); depending on their needs (local run, external platform, curated, or non-curated GSMNs...), users can choose either method and connect it to M2M.

Most metabolic modelling methods rely on flux analyses (*Orth et al., 2010*) solved with linear programming, which may turn out to be challenging to implement for simulations of large communities (*Basile et al., 2020*), although recent efforts in that direction are encouraging (*Popp and Centler, 2020*). M2M uses the network expansion algorithm and solves combinatorial optimisation problems with Answer Set Programming, thereby ensuring fast simulations and community predictions, suitable when performing systematic screening and multiple experiments. Network expansion has been widely used to analyse and refine metabolic networks (*Matthäus et al., 2008*; *Laniau et al., 2017*; *Christian et al., 2009*; *Prigent et al., 2017*), including for microbiota analysis (*Christian et al., 2007*; *Ofaim et al., 2017*; *Opatovsky et al., 2018*; *Frioux et al., 2018*). Network expansion is a complementary alternative to quantitative constraint-based methods (*Ebenhöh et al., 2004*; *Handorf et al., 2005*) such as flux balance analysis as it does not require biomass reactions nor accurate stoichiometry. This algorithm offers a good trade-off between the accuracy of metabolic predictions and the precision required for the input data, adapted to the challenges in studying non-model organisms and their likely incomplete models of metabolism (*Bernstein et al., 2019*).

Answer Set Programming can easily scale the analysis of minimal communities among thousands of networks considered in interaction and ensures with efficient solving heuristics that the whole space of solutions is parsed to retrieve key species for chosen end-products. M2M therefore suggests (metagenomic) species for further analyses such as targeted curation of metabolic networks and deeper analysis of the genomes or quantitative flux predictions. The relevance of MiSCoTo (*Frioux et al., 2018*), the algorithm for minimal community selection used in M2M, has been recently experimentally demonstrated. It was applied to design bacterial communities to support the growth of a brown alga in nearly axenic conditions (*Burgunter-Delamare et al., 2019*). Despite the difficulty inherent to controlling the communities for a complex alga, the inoculated algae exhibited a significant increase in growth and metabolic profiles that at least partially aligned with the predictions, demonstrating the versatility in application fields of our methods.

There are limitations associated to the software solution described in this paper. One challenge in applying our tool is to accurately estimate the nutrients available in a given environment (seeds), on which the computation of network expansion relies. The algorithm provides a snapshot of producible metabolites, representing the sub-network that can be activated under given nutritional conditions. However, network expansion has been shown to be sensitive to cycles in GSMNs and it is therefore relevant to include some cofactors (or currency metabolites e.g. ADP) in the seeds to activate such cycles, the way many studies proceed (*Cottret et al., 2010*; *Greenblum et al., 2012*; *Eng and Borenstein, 2016*; *Julien-Laferrière et al., 2016*). In addition, it has to be noted that the cost of exchanges or their number are not taken into account in M2M. Transport reactions are hardly recovered by automatic methods (*Bernstein et al., 2019*) and validation of cross-feedings implies an additional work on transporters identification. The standalone MiSCoTo package used in M2M has a solving mode taking into account exchanges: it can compute communities while minimising and suggesting metabolic exchanges, although this comes with additional computational costs and a need for validation. Another limitation of our approach for studying communities of individuals from

metagenomic experiments is that we do not take the microbial load or abundance of MGS into account in the pipeline. Considering the presence/absence of MGS might lead to overestimate the production of some metabolites. In addition, we infer phenotypes directly from genotypes, thereby ignoring the possible non-expression of metabolic-related genes in specific conditions and the regulation of those genes. Metaproteomic and metatranscriptomic data could partly overcome these shortcomings but such experiments are not yet routinely performed. Finally, another aspect to be considered in the future is the competition between species, especially for nutrients, as we only focus here on metabolic complementarity and positive interactions.

Despite the above mentioned limitations, M2M has multiple applications for the de novo screening of metabolism in microbial communities. The number of curated GSMNs for species found in microbiotas increases (*Magnúsdóttir et al., 2017*), constituting a highly valuable resource for the study of interactions by mapping metagenomic data or OTUs to the taxonomy of genomes associated to these GSMNs. Yet, the variety of (reference) genomes obtained from shotgun metagenomic experiments is such than species and strains may not belong to the ones for which a curated GSMN is available. In that case, the proportion of reads that are not mapped to a genome with an associated GSMN can be very high (*Diener et al., 2020*). In addition, predictions from GSMN mapping can be misleading as it is known that genomes vary a lot between genera, species, and even strains, (*Ansorge et al., 2019*) and so can the metabolism. Recent methods for assembling genomes directly from metagenomes lead to nearly complete genomes for possibly unknown species on which one may still want to get metabolic insights (*Almeida et al., 2019*). Long-reads sequencing associated with short-reads sequencing can also give access to complete microbial genomes (*Moss et al., 2020*). Finally, single-cell methods can be useful for the acquisition of genomes and metagenomes (*Treitli et al., 2019*). M2M answers to the need for de novo metabolic inference and screening, which is likely to become a routine in the rapidly evolving context of microbiota genome sequencing. While studying the metabolic potential of large communities is an iterative process that still requires biological expertise, we provide with this work means to facilitate the screening of metagenomes and reduce these large communities to key members.

## Conclusion

M2M allows metabolic modelling of large-scale communities, based on reference genomes or de novo constructed MAGs, inferring metabolic complementarity found within communities. M2M is a flexible framework that automates GSMN reconstruction, individually and collectively analyse GSMNs, and performs community selection for targeted functions. The large combinatorics of minimal communities due to functional redundancy in microbiotas is addressed by providing key species associated to metabolic end-products. This could allow targeting specific members of the community through pro- or prebiotics, to model the metabolites the human host will be exposed to.

We validated the flexibility of the software and the range of analyses it can offer with several datasets, corresponding to multiple use-cases in the microbiome field. This allowed us to characterise metabolic complementarity in a large collection of draft reference genomes. We further assessed the robustness of M2M to data incompleteness by performing analyses on collections of MAGs. Finally, we applied M2M to a common use-case in metagenomics: the study of communities associated to individuals, in a disease context.

Our method is robust against the uncertainty inherent to metagenomics data. It scales to typical microbial communities found in the gut and predicts key species for functions of interest at the metabolic level. Future developments will broaden the range of interactions to be modelled and facilitate the incorporation of abundance data. This software is an answer to the need for scalable predictive methods in the context of metagenomics where the number of available genomes continues to rise.

## Materials and methods

M2M is a Python package. It can be used on a workstation or on a cluster using Docker or Singularity. M2M's source code is available on github.com/AuReMe/metage2metabo, (*Belcour and Frioux, 2020*; copy archived at swh:1:rev:2cab4c79acd814eb177a370602c07599a93bc947) and the package is available though the Python Package Index at pypi.org/project/Metage2Metabo/. A detailed documentation is available on metage2metabo.readthedocs.io.

We detail below the characteristics of M2M through a description of its main steps.

## Parallel and large-scale metabolic network reconstruction

M2M can process existing metabolic networks in SBML format or proposes the automatic reconstruction of non-curated metabolic networks (m2m recon). As a multi-processing solution, it facilitates the treatment of hundreds or thousands of genomes that can be retrieved from metagenomic experiments. The underlying GSMN reconstruction software is Pathway Tools (*Karp et al., 2016*), a graphical user interface (GUI) based software suite for the generation of individual GSMNs, called Pathway/Genome Databases (PGDBs). Typically, a PGDB is obtained from an annotated genome using PathoLogic, the software prediction component of Pathway Tools, and curated afterwards.

We developed Mpwt (Multiprocessing Pathway Tools), a command-line Python wrapper (also available as a standalone tool) for Pathway Tools. Mpwt and M2M (i) format the genomic inputs, (ii) automate the reconstruction step by initialising a PathoLogic environment for each genome, and (iii) extract and convert the resulting GSMNs in PGDB and SBML (*Hucka et al., 2003*; *Hucka et al., 2018*) formats using the PADMet library (*Aite et al., 2018*). Mpwt handles three types of genomic inputs (Genbank, Generic Feature Format (GFF) or PathoLogic format) that must contain GO-terms and EC-numbers annotations necessary for Pathway Tools. These annotations are for example found in the Genbank files generated by Prokka (*Seemann, 2014*). In addition, we specifically developed Emapper2gbk, a Python package dedicated to the connection between the Eggnog-mapper annotation tool (*Huerta-Cepas et al., 2017*) and Mpwt in order to generate these inputs.

## Analysis of metabolic producibility and calculation of the cooperation potential

This part of the workflow encompasses three steps: computation of the (i) individual (m2m iscope) and (ii) collective (m2m cscope) metabolic potentials, and (iii) the characterisation of the cooperation potential of the GSMN collection (m2m addedvalue). The former two rely on the network expansion algorithm (*Ebenhöh et al., 2004*), the latter being a set difference between the results of the first two steps.

The network expansion algorithm computes the *scope* of a metabolic network from a description of the growth medium called *seeds*. The scope consists in the set of metabolic compounds which are reachable, or producible, according to a boolean abstraction of the network dynamics assuming that cycles cannot be self-activated. More precisely, the algorithm recursively considers products of reactions to be producible if all reactants of the reactions are producible, provided an initiation with a set of seed nutrients. The underlying implementation of the network expansion algorithm used in M2M relies on Answer Set Programming (ASP) (*Schaub and Thiele, 2009*).

We define a metabolic network as a bipartite graph $G = (R \cup M, E)$, where $R$ and $M$ stand for reaction and metabolite nodes. When $(m, r) \in E$ (respectively $(r, m) \in E$), with $m \in M$ and $r \in R$, the metabolite is called a *reactant* (respectively *product*) of the reaction $r$. The scope of a set of seed compounds $S$ according to a metabolic network $G$, denoted by $(G, S)$, is iteratively computed until it reaches a fixed point (*Handorf et al., 2005*). It is formally defined by

$$\text{Scope}(G, S) = \bigcup_i M_i, \text{ where } M_0 = S \text{ and } M_{i+1} = M_i \cup products(\{r \in R \mid reactants(r) \subseteq M_i\}).$$

### Individual metabolic capabilities

The m2m iscope command predicts the set of reachable metabolites for each GSMN using the network expansion algorithm and the given nutrients as seeds. The content of each scope is exported to a json file. A summary is also provided to the user comprising the intersection (metabolites reachable by all GSMNs) and the union of all scopes, as well as the average size of the scopes, the minimal size and the maximal size of all. This command extends core functions implemented in Menetools for individual GSMNs, a Python package (also available as a standalone tool) that was previously used in *Aite et al., 2018*.

### Collective metabolic capabilities

The m2m cscope command computes the metabolic capabilities of the whole microbiota by taking into account the metabolic complementarity between GSMNs. This step simulates the sharing of

metabolic biosynthesis through a meta-organism composed of all GSMNs, and assesses the metabolic compounds that can be reached using network expansion. This calculation is an extension of the features of MiSCoTo (also available as a standalone tool) (*Frioux et al., 2018*) in which the collective scope of a collection of metabolic networks $\{G_1, \ldots G_N\}$ is introduced. We define

$$\text{collectiveScope}(G_1...G_N, S) = \text{Scope}\left(\left(\bigcup_{i \in \{1...n\}} R_i, \bigcup_{i \in \{1...n\}} M_i, \bigcup_{i \in \{1...n\}} E_i\right), S\right).$$

## Target producers

If metabolic compounds of interest or *targets* are provided by the user, a summary of the producers for each target is generated by `m2m workflow`, `m2m metacom`, and `m2m cscope`: it identifies the GSMNs that are predicted to produce the targets, either intrinsically, or through cooperation with other members of the community.

A metabolic network $G_i$ is an *individual target producer* of $t \in T$ if $t \in (G_i, S)$. The metabolic network $G_i$ is a *community target producer* if (*a*) $G_i$ is not an individual target producer of $t$ (i.e. $t \notin (G_i, S)$), but (*b*) $G_i$ contains a reaction $r \in R_i$ which produces $t$ (i.e. $t \in products(r)$) such that (*c*) all reactants are producible by the community ($G_i$ and the other metabolic networks): $reactants(r) \subset \text{collectiveScope}(G_1...G_N, S)$. This means that the metabolic network $G_i$ has the capability of producing $t$ through the reaction $r$ in a cooperation context.

This information can be retrieved in practice in the file 'producibility_targets.json' under the keys 'individual_producers' and 'com_only_producers'.

## Cooperation potential

Given individual and community metabolic potentials, the *cooperation potential* consists in the set of metabolites whose producibility can only occur if several organisms participate in the biosynthesis. `m2m addedvalue` computes the cooperation potential by performing a set difference between the community scope and the union of individual scopes, and produces an SBML file with the resulting metabolites. This list of compounds is inclusive and could comprise false positives not necessitating cooperation for production, but selected due to missing annotations in the initial genomes. One can modify the SBML file accordingly, prior to the following M2M community reduction step.

The cooperation potential $(G_1, ..., G_n, S)$ of a collection of metabolic networks $\{G_1...G_n\}$ is defined by

$$\text{cooperationPotential}(G_1, ..., G_n, S) = \text{collectiveScope}(G_1, ..., G_n, S) \setminus \bigcup_{i \in \{1...n\}} \text{scope}(G_i, S).$$

# Computation of minimal communities and identification of key species

A minimal community $\mathcal{C}$ enabling the producibility of a set of targets $T$ from the seeds $S$ is a sub-family of the community $G_1, \ldots, G_n$ which is solution of the following optimisation problem:

$$\underset{\{G_{i_1}...G_{i_L}\} \subset \{G_1...G_N\}}{\text{minimize}} \quad \text{size}(\{G_{i_1}...G_{i_L}\})$$
$$\text{subject to} \quad T \subset \text{collectiveScope}(G_{i_1}...G_{i_L}, S).$$

Solutions to this optimisation problem are communities $\mathcal{C} = (G_{i_1} \ldots, G_{i_L})$ of minimal size. We define $\text{minimalCommunities}(G_1...G_n, S, T)$ to be the set of all such minimal communities. A first output of the `m2m mincom` command is the (minimal) size $L$ of communities solution of the optimisation problem. The composition of one optimal community is also provided. The targets are by default the components of the cooperation potential, $T = \text{cooperationPotential}(G_1, ..., G_n, S)$, but can also be a group of target metabolites defined by the user.

Many minimal communities are expected to be equivalent for a given metabolic objective but their enumeration can be computationally costly. We define *key species* which are organisms occurring in at least one community among all the optimal ones. Key species can be further distinguished into *essential symbionts* and *alternative symbionts*. The former occur in every minimal community whereas the latter occur only in some minimal communities. More precisely, the key species $\text{keySpecies}(G_1...G_n, S, T)$, the essential symbionts $\text{essentialSymbionts}(G_1...G_n, S, T)$, and the alternative

symbionts $\mathrm{alternativeSymbionts}(G_1...G_n, S, T)$ associated to a set of metabolic networks, seeds $S$ and a set of target metabolites $T$ are defined by

$$\begin{aligned}
\mathrm{keySpecies}(G_1...G_n, S, T) &= \{G \mid \exists \mathcal{C} \in \mathrm{minimalCommunities}(G_1...G_n, S, T), G \in \mathcal{C}\}. \\
\mathrm{essentialSymbionts}(G_1...G_n, S, T) &= \{G \mid \forall \mathcal{C} \in \mathrm{minimalCommunities}(G_1...G_n, S, T), G \in \mathcal{C}\}. \\
\mathrm{alternativeSymbionts}(G_1..G_n, S, T) &= \mathrm{keySpecies}(G_1...G_n, S, T) \setminus \mathrm{essentialSymbionts}(G_1...G_n, S, T).
\end{aligned}$$

As a strategy layer over MiSCoTo, M2M relies on the Clasp solver (*Gebser et al., 2012*) for efficient resolution of the underlying grounded ASP instances. Although this type of decision problem is NP-hard (*Julien-Laferrière et al., 2016*), as with many real-world optimisation problems worst-case asymptomatic complexity is less informative for applications than practical performance using heuristic methods. The Clasp solver implements a robust collection of heuristics (*Gebser et al., 2007*; *Andres et al., 2012*) for core-guided weighted MaxSAT (*Manquinho et al., 2009*; *Morgado et al., 2012*) that provide rapid set-based solutions to combinatorial optimisation problems, much in the same way that heuristic solvers like CPlex provide rapid numerical solutions to mixed integer programming optimisation problems. The kinds of ASP instances constructed by MiSCoTo for M2M are solved in a matter of minutes for the identification of key species and essential/alternative symbionts. Indeed the space of solutions is efficiently sampled using adequate projection modes in ASP, which enables the computation of these groups of species without the need for a full enumeration.

### Analysis of enumerated communities

The `m2m_analysis` command permits the enumeration of minimal communities. If the taxonomy of species associated to the metabolic networks is provided, descriptive statistics are performed. In addition, minimal communities can be visualised as an association graph connecting GSMNs that co-occur in at least one minimal community. The association graph can itself be compressed in a power graph that enables visualising motifs such as cliques, bicliques and stars. Power graphs are generated using PowerGrASP (*Bourneuf and Nicolas, 2017*). In this paper, they were visualised with Cytoscape (v.2.8.3) (*Shannon et al., 2003*) and the CyOog plugin (v.2.8.2) developed by *Royer et al., 2008*.

## Application to datasets

### Analysis of human gut and cow rumen published collections of genomes

In order to evaluate the influence of genome collections based on sequencing cultured isolates or metagenomic genome reconstructions, we used 1,520 high-quality draft reference genomes of bacteria from the human gut microbiota retrieved from *Zou et al., 2019* and 913 MAGs from the cow rumen published in *Stewart et al., 2018*. The genomes from the former set were already annotated.

We designed a set of seed metabolites representing a nutritional environment which is required for the metabolic modelling analyses. Seeds (93 metabolites) consist in components of a classical diet for the gut microbiota, EU average from the VMH resource (*Noronha et al., 2019*), and a small number of currency metabolites (*Schilling et al., 2000*; *Supplementary file 1* - Table 1 and Github repository of M2M). M2M was run using version 23.0 of Pathway Tools.

The cow rumen dataset of MAGs was not functionally annotated. Therefore, as a preliminary step of analysis, we annotated the genomic contigs using Prokka (v.1.13.4) (*Seemann, 2014*). M2M was run using version 23.0 of Pathway Tools. The nutritional environment for modelling experiments consisted in basic nutrients: 26 metabolites including inorganic compounds, carbon dioxide, glucose, and cellobiose and a small number of currency metabolites (*Supplementary file 1* - Table 2).

The rumen MAGs were artificially degraded to assess the robustness of M2M with respect to incomplete MAGs. This was done by randomly removing genes in all or a fraction of genomes. Four degradation scenarios were tested: removal of 2% of genes in all MAGs, removal of 5% of genes in 80% of the genomes, removal of 5% of genes in all genomes and removal of 10% of genes in 70% of the genomes. The subsequent parts of the analysis (annotation with Prokka, M2M runs) were done as described above. Supervenn diagrams presented in *Figure 2* to compare the results were obtained using the Supervenn Python package (*Fedor, 2021*).

## Shotgun metagenomic analysis of individuals

Metagenomic shotgun data from samples previously studied in *Diener et al., 2020* from 186 Danish and Swedish individuals (*Forslund et al., 2015*) were used in this paper. Genomes were de novo reconstructed from the dataset using the MATAFILER pipeline described in *Hildebrand et al., 2019*. Briefly, metagenomic samples were quality-filtered using sdm (*Hildebrand et al., 2014*), assembled using MEGAHIT (*Li et al., 2015*), genes were predicted using Prodigal (*Hyatt et al., 2010*), and a non-redundant gene catalogue was constructed across all samples using MMseqs2 (*Steinegger and Söding, 2017*). MAGs were predicted from metagenomic assemblies using MetaBAT2 (*Kang et al., 2019*) and dereplicated into species level metagenomic species (MGS), using a combination of shared genes among MetaBAT2 bins, canopy clustering (*Nielsen et al., 2014*) and custom R scripts (*Hildebrand et al., 2019*). Abundance of MGS was estimated across samples by using the average coverage of 40 conserved, single copy marker genes associated to each MGS (*Mende et al., 2013*). This abundance matrix was further populated with specI species from the proGenomes database (*Mende et al., 2020*), that were not represented by MGS and are high-quality genomes from cultured bacteria. This pipeline is described in further detail in *Hildebrand et al., 2019*.

Samples for which the global estimated abundance of MGS was lower than 1000 in accumulated coverage of all species were removed. This corresponds to a low number of reads passing the upstream quality checks for these samples (<9.10e6). 170 samples were kept for analysis. The initial bacterial community of each sample was determined using the estimated abundance matrix provided by the MATAFILER pipeline, following a boolean rule of presence/absence of MGS and specI species in samples. Genomes consisted in MGS obtained with MATAFILER as well as SpecI genomes from the Progenomes database (*Mende et al., 2020*) that were identified in the samples. For the latter case, we downloaded the genes and proteins of the corresponding representative genome from the database (SpecI v3). Functional annotation of genes from both specI genomes and MGS core genomes was performed using EggNOG-mapper v2.0.0 (*Huerta-Cepas et al., 2017*) based on eggNOG orthology data (*Huerta-Cepas et al., 2019*). Sequence searches were performed using Diamond v0.9.24.125 (*Buchfink et al., 2015*). Treatment of EggNOG-mapper annotation and creation of M2M Genbank inputs was done with the package Emapper2gbk that we developed for the project. Seeds describing the nutritional environment were compounds of the western diet as presented in the study by *Diener et al., 2020*. These metabolites were translated into identifiers from the Metacyc database (*Caspi et al., 2020*), to which were added a small number of currency metabolites. The seeds are available on the Github repository of M2M.

M2M was run for each sample and community selection was performed with different sets of targets (SCFAs, cooperation potential in each community). The cohort and disease status of each sample was known, enabling the comparison of scopes and cooperation potentials contents between statuses. M2M was also run using the approach presented in MICOM (*Diener et al., 2020*) building sample communities, as presented in the paper and associated data repository, through the attribution of curated GSMN (*Magnúsdóttir et al., 2017*) to operational taxonomic units (OTUs) identified in samples. We reused the mapping at species-level provided in the MICOM paper to build the communities.

Downstream analyses were performed in R (*R Development Core Team, 2017*) and Python. Figures were produced using the package ggplot2 (*Wickham, 2009*) and diversity measures were computed with the vegan R package (v2.5–6). Classifications of disease statuses or cohorts using sets of predicted producible metabolites were made using the Python package Scikit-learn (v0.23.1) (*Pedregosa et al., 2011*). Briefly, redundancy between features were removed with a Multidimensional scaling (MDS), Support Vector Machine (SVM) classifications with Stratified K-Folds cross-validations were performed using the MDS results. Finally, receiver operating characteristic curve (ROC-AUC) were computed and visualised with tools from the package.

## Acknowledgements

The authors acknowledge the GenOuest bioinformatics core facility for providing the computing infrastructure. This research was supported in part by the NBI Computing infrastructure for Science (CiS) group. FH and CF's salaries have been funded by the BBSRC Institute Strategic Programme

Gut Microbes and Health BB/r012490/1, its constituent project BBS/e/F/000Pr10355. AB's, CF,and MA's salaries have been funded by the ANR project IDEALG (ANR-10-BTBR-04) 'Investissements d'Avenir, Biotechnologies-Bioressources'. The authors acknowledge P Karp, S Paley, M Krummenacker, R Billington, A Kothari from the Bioinformatics Research Group of SRI International for their help regarding Pathway Tools. The authors also thank Lucas Bourneuf for his help on power graph analyses, Yann Le Cunff for his help regarding statistical analyses, and Samuel Blanquart and David Sherman for their useful comments on the manuscript.

## Additional information

### Funding

| Funder | Grant reference number | Author |
|--------|------------------------|--------|
| BBSRC | Gut Microbes and Health BB/r012490/1 and its constituent project BBS/e/F/000Pr1035 | Clémence Frioux<br>Falk Hildebrand |
| ANR | IDEALG (ANR-10-BTBR-04) "Investissements d'Avenir. Biotechnologies-Bioressources" | Arnaud Belcour<br>Clémence Frioux<br>Méziane Aite |

The funders had no role in study design, data collection and interpretation, or the decision to submit the work for publication.

### Author contributions

Arnaud Belcour, Conceptualization, Data curation, Software, Validation, Investigation, Visualization, Methodology, Writing - review and editing; Clémence Frioux, Conceptualization, Data curation, Software, Supervision, Validation, Investigation, Visualization, Methodology, Writing - original draft, Writing - review and editing; Méziane Aite, Anthony Bretaudeau, Software; Falk Hildebrand, Resources, Data curation, Software, Validation, Writing - review and editing; Anne Siegel, Conceptualization, Supervision, Funding acquisition, Writing - review and editing

### Author ORCIDs

Arnaud Belcour https://orcid.org/0000-0003-1170-0785
Clémence Frioux https://orcid.org/0000-0003-2114-0697
Méziane Aite https://orcid.org/0000-0001-9086-1485
Anthony Bretaudeau https://orcid.org/0000-0003-0914-2470
Falk Hildebrand https://orcid.org/0000-0002-0078-8948
Anne Siegel https://orcid.org/0000-0001-6542-1568

### Decision letter and Author response

Decision letter https://doi.org/10.7554/eLife.61968.sa1
Author response https://doi.org/10.7554/eLife.61968.sa2

## Additional files

### Supplementary files

- Supplementary file 1. Supplementary tables 1 to 25.
- Transparent reporting form

### Data availability

Source code and data related to this article are available in https://github.com/AuReMe/metage2-metabo/tree/master/article_data (copy archived at https://archive.softwareheritage.org/swh:1:rev:2cab4c79acd814eb177a370602c07599a93bc947/).

The following previously published datasets were used:

| Author(s) | Year | Dataset title | Dataset URL | Database and Identifier |
|---|---|---|---|---|
| Zou Y, Xue W, Luo G, Deng Z, Qin P, Guo R | 2018 | Genome Sequencing and Assembly for Cultivated Species of the Human Gut Microbiota | https://www.ncbi.nlm. nih.gov/bioproject/ PRJNA482748 | NCBI BioProject, PRJNA482748 |
| Stewart RD, Auffret MD, Warr A, Wiser AH, Press MO, Langford KW | 2017 | Metagenomic sequencing of the rumen of Scottish cattle | https://www.ncbi.nlm. nih.gov/bioproject/? term=PRJEB21624 | NCBI BioProject, PRJEB21624 |
| Karlsson FH, Tremaroli V, Nookaew I, Bergström G, Behre CJ, Fagerberg B, Nielsen J, Bäckhed F | 2013 | Gut metagenome in European women with normal, impaired and diabetic glucose control | https://www.ebi.ac.uk/ ena/browser/view/ PRJEB1786 | European Nucleotide Archive, PRJEB1786 |
| Li J, Jia H, Cai X, Zhong H, Feng Q, Sunagawa S, Arumugam M, Kultima JR, Prifti E, Nielsen T, Juncker AS, Manichanh C, Chen B, Zhang W, Levenez F, Wang J, Xu X, Xiao L, Liang S, Zhang D, Zhang Z, Chen W, Zhao H, Al-Aama JY, Edris S, Yang H, Hansen T, Nielsen HB, Brunak S, Kristiansen K, Guarner F, Pedersen O, Doré J, Ehrlich SD, MetaHIT Consortium, Bork P, Wang J | 2013 | Gut metagenome in European women with normal, impaired and diabetic glucose control | https://www.ebi.ac.uk/ ena/browser/view/ PRJEB5224 | European Nucleotide Archive , PRJEB5224 |
| Le Chatelier E, Nielsen T, Qin J, Prifti E, Hildebrand F, Falony G, Almeida M, Arumugam M, Batto JM, Kennedy S, Leonard P, Li J, Burgdorf K, Grarup N, Jørgensen T, Brandslund I, Nielsen HB, Juncker AS, Bertalan M, Levenez F, Pons N, Rasmussen S, Sunagawa S, Tap J, Tims S, Zoetendal EG, Brunak S, Clément K, Doré J, Kleerebezem M, Kristiansen K, Renault P, Sicheritz-Ponten T, de Vos WM, Zucker JD, Raes J, Hansen T, MetaHIT consortium, Bork P, Wang J, Ehrlich SD, Pedersen O | 2013 | Richness of human gut microbiome correlates with metabolic markers | https://www.ebi.ac.uk/ ena/browser/view/ PRJEB4336 | European Nucleotide Archive , PRJEB4336 |

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

## Appendix 1

### Analysis of GSMNs from the human gut reference genomes and rumen MAGs collections

#### Comparison of GSMNs reconstructed from MAGs and from reference genomes

Main characteristics of GSMNs reconstructed from the human gut microbiota reference genomes and the rumen MAGs are compared in *Appendix 1—figure 1*.

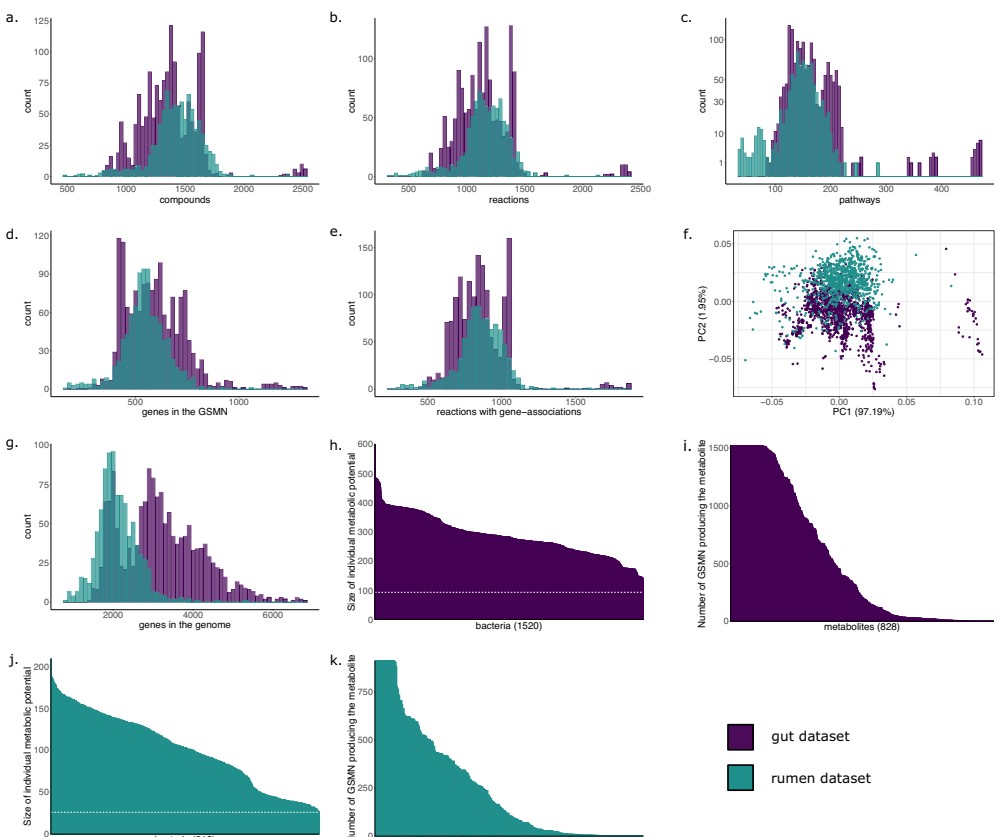

**Appendix 1—figure 1.** Characteristics of the metabolic networks built for the gut and the rumen datasets. (**a**) Distribution of the number of metabolic compounds in genome-scale metabolic networks (GSMNs) reconstructed for the gut dataset (purple) and the rumen dataset (green). (**b**) Distribution of the number of metabolic reactions. (**c**) Distribution of the number of complete pathways according to the MetaCyc database. (**d**) Distribution of the number of genes included into the GSMNs. (**e**) Distribution of the number of reactions associated to genes. (**f**) Principal component analysis of the GSMNs reconstructions based on the previous characteristics (a. to e.). (**g**) Distribution of the number of genes (not necessarily related to metabolism) in the initial genomes/MAGs. (**h**) Individual metabolic potentials (scopes) for the gut bacteria, dotted line represents the number of seeds (nutrients) used in the algorithm. (**i**) Reachability of metabolites by gut bacteria. (**j**) Individual metabolic potentials (scopes) for the rumen bacteria, dotted line represents the number of seeds (nutrients) used in the algorithm. (**k**) Reachability of metabolites by rumen bacteria.

#### Robustness analysis of GSMN reconstruction with MAGs

MAGs from the rumen dataset were degraded by randomly removing contigs. The following degradations were tested: removal of 2% of genes in all MAGs, removal of 5% of genes in 80% of MAGs, removal of 5% of genes in all MAGs, removal of 10% of genes in 70% of MAGs.

*Appendix 1—table 1* summarises the characteristics of the genomes and GSMNs for all experiments. The average gene loss in genomes is similar to the average gene loss in metabolic networks. However, the average loss of metabolites and reactions is lower than the genetic loss: it increases more slowly than the loss of genes. For instance, the 2% degradation of MAGs leads to a nearly 2% decrease in reaction numbers in GSMNs. However, the 10% degradation in 70% of genomes (average gene loss of 7% in the initial community) only leads to a 5% decrease in reaction numbers. One notable observation is the stability in the percentage of reactions associated to genes, suggesting that the loss of reactions in degraded genomes mainly occurs among reactions that are not associated to genes. It is also possible that the loss of genes in GSMNs is due to the redundancy loss: some reactions associated to several genes before degradation lose some of these gene associations after degradation. Data for each genome and GSMN is available in *Supplementary file 1* - Tables 16, 21-24.

**Appendix 1—table 1.** Effect of MAG degradation on genome-scale metabolic network (GSMN) reconstructions.

Numbers are averages. '±' precedes standard deviation values. 'original': initial MAGs prior degradation, '2pc100': 2% gene removal in all MAGs, '5pc80': 5% gene removal in 80% of MAGs, '5pc100': 5% gene removal in all MAGs, '10pc70': 10% gene removal in 70% of MAGs.

| | Original | 2pc100 | 5pc80 | 5pc100 | 10pc70 |
|---|---|---|---|---|---|
| Genes in MAGs | 2100 (±501) | 2058 (±491) | 2016 (±484) | 1994 (±478) | 1954 (±480) |
| Reactions in GSMNs | 1155 (±199) | 1131 (±192) | 1116 (±192) | 1108 (±190) | 1094 (±192) |
| Metabolites in GSMNs | 1422 (±212) | 1402 (±207) | 1388 (±208) | 1381 (±206) | 1366 (±208) |
| Genes in GSMNs | 543 (±108) | 532 (±106) | 521 (±105) | 515 (±103) | 505 (±105) |
| % reactions with genes | 73.84% | 74.05% | 73.82% | 73.72% | 73.61% |
| Gene loss in MAGs | — | 1.98% | 4.01% | 5.03% | 6.94% |
| Reaction loss in GSMNs | — | 1.96% | 3.30% | 3.89% | 5.17% |
| Metabolite loss in GSMNs | — | 1.37% | 2.41% | 2.91% | 3.92% |
| Gene loss in GSMNs | — | 2.09% | 4.17% | 5.11% | 7.02% |

## Appendix 2

### Supplementary information to the Diabetes experiment
#### Diversity and richness of the samples

The Shannon diversity index and richness of the 170 samples are illustrated in *Appendix 2—figure 1* a and b. We relied on species present in the abundance matrix to define communities for the metabolic analysis. The phylum-level composition of the matrix is illustrated in *Appendix 2—figure 2*. The average size of the community was 108 GSMNs. Their median size was 111. In order to compute a metabolic distance between samples, we retrieved for each genome its KO annotations obtained with Eggnog-Mapper. Using the abundance (normalised by sample) of the genomes in each sample, we were able to retrieve the KO content of samples. We then calculated the Bray-Curtis distance between samples before computing a PCoA (*Appendix 2—figure 1*). The PCoA shows a clear distinction between the two datasets, thus motivating their distinct analysis, as performed in the main results of the article. However, there are no distinction between the control and diabetes status of the samples.

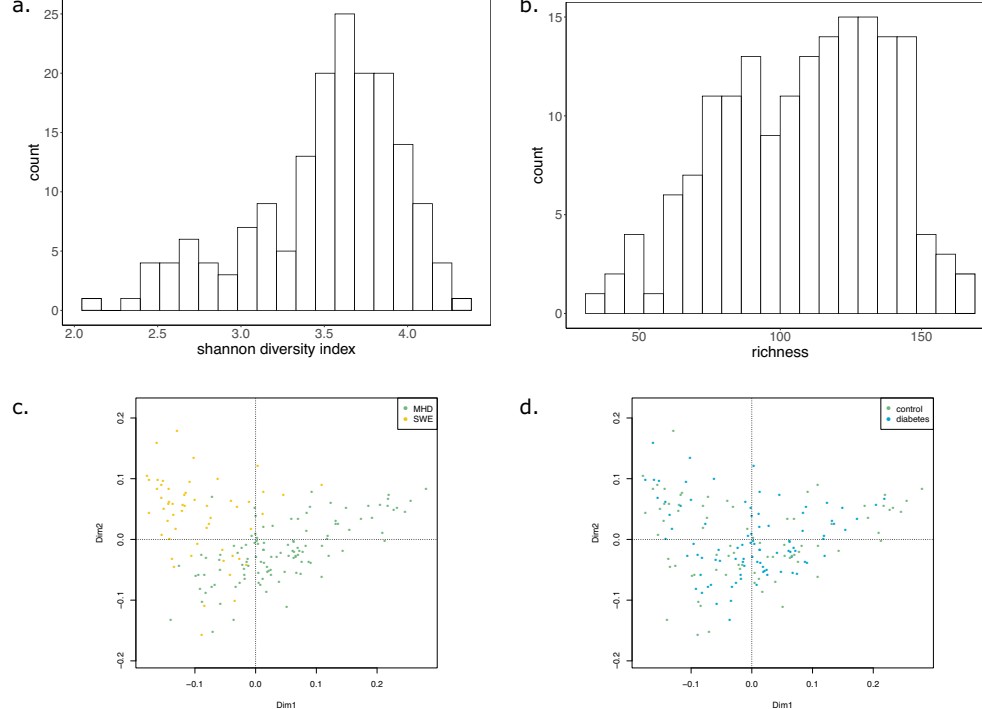

**Appendix 2—figure 1.** Shannon diversity index, richness, and metabolic distance of the samples. (**a**) Histogram depicting the Shannon diversity index of the samples. (**b**) Histogram depicting the richness of the samples. (**c** and **d**) Principal coordinate analysis (PCoA) of the Bray-Curtis distance calculated on the KO composition of samples coloured by dataset (**c**) or disease status (**d**).

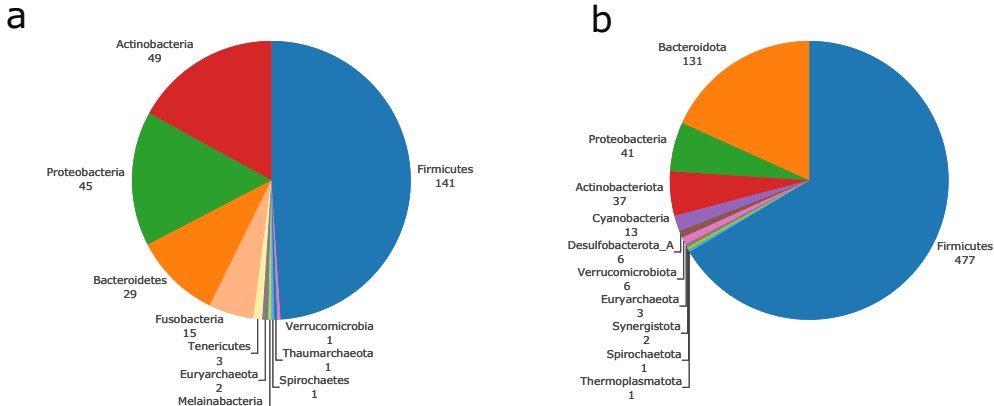

**Appendix 2—figure 2.** Taxonomic diversity of the genomes used for genome-scale metabolic networks (GSMNs) reconstruction using OTU mapping (at species level) to curated metabolic models (a), or reconstructed metagenomic species (MGS) (b). Phyla composition of the genomes, and number of distinct representatives for each phylum.

## Cohort effect at the metabolic level

The cohort effect is illustrated in *Appendix 2—figure 3*.

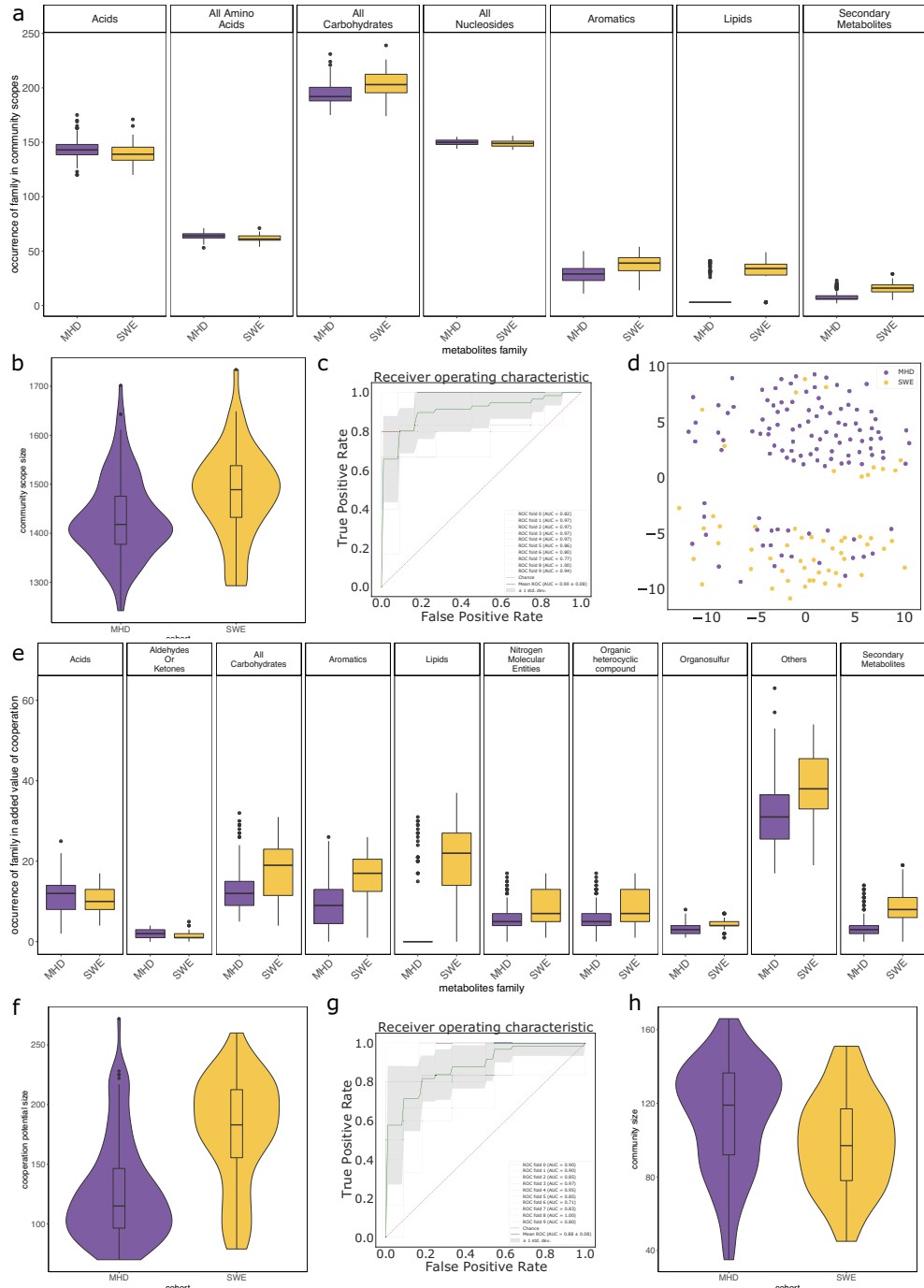

**Appendix 2—figure 3.** Impact of the cohort when studying the metabolisms of individuals from the metagenomic dataset. Panels (**a**) to (**d**) focus on the community scope, that is the set of metabolites reachable by the community associated to a sample. Panel (**d**) shows the representation of a multidimensional scaling (MDS) on the community scope composition between cohorts. Panels (**e**) to (**g**) focus on the cooperation potential, that is the set of metabolites that are not expected to be produced by individual members of communities and instead require cooperation. Panels (**a**) and (**e**) describe families of metabolites whose occurrences significantly differ between cohorts in the corresponding group (community scope or cooperation potential). Panels (**b**) and (**f**) illustrate the size of the community scope and cooperation potential respectively in samples from the two cohorts. Panels **c** (resp. **g**) are receiving operating curves (ROC) of a classification experiment aiming

*Appendix 2—figure 3 continued*

at separating the cohort (MHD n = 115, SWE n = 55) based on the occurrences of metabolites in the community scope (resp. cooperation potential). Panel (**h**) describes the size of the initial community associated to samples of both cohorts according to abundance data of MGS.

## Status effect at the metabolic level

The effect of the disease status is illustrated in *Appendix 2—figure 4*.

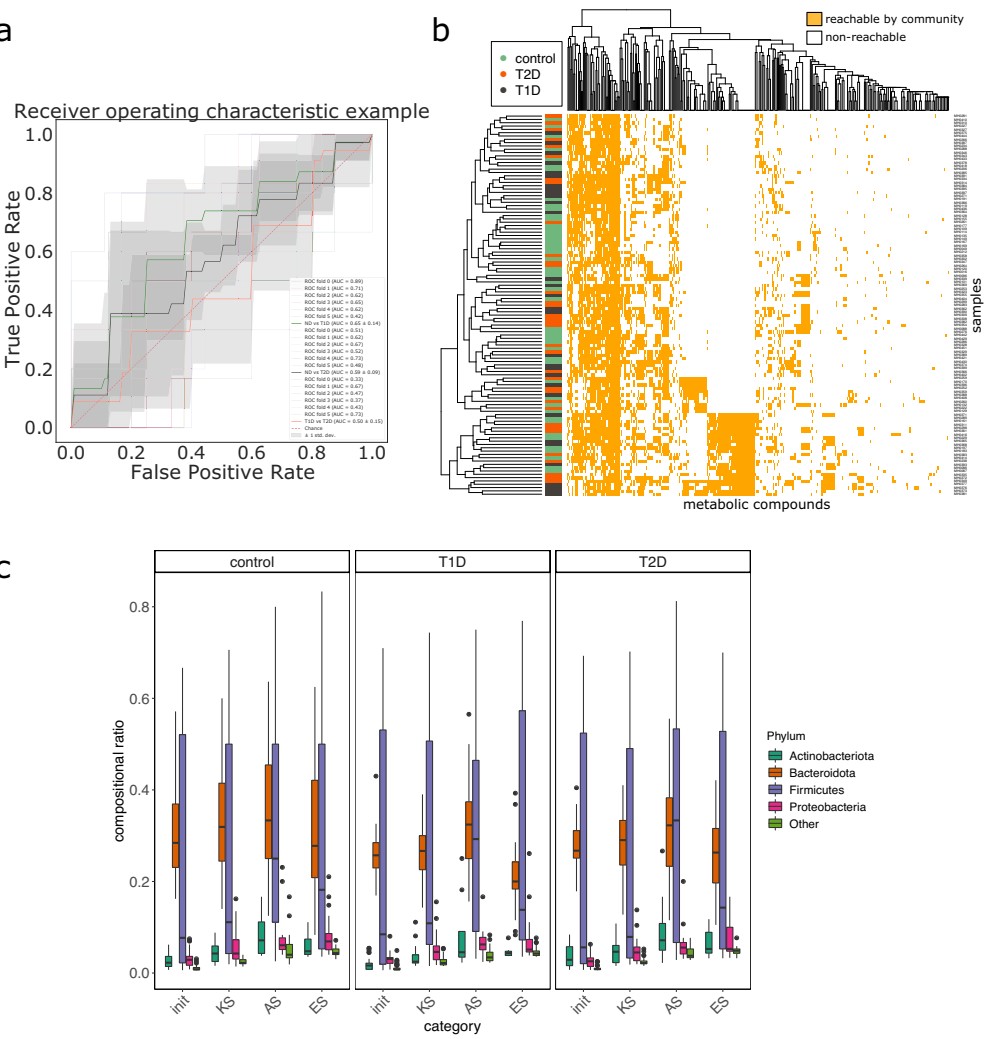

**Appendix 2—figure 4.** Impact of the status when studying the metabolisms of individuals from the MHD metagenomic dataset. Panel (**a**) is the receiver operating curve (ROC) of the classification experiment aiming at deciphering the disease status for the MHD cohort (control n = 49, Type-1 Diabetes n = 31 or Type-2 Diabetes n = 35) based on the cooperation potential composition. Panel (**b**) illustrates the cooperation potential composition in terms of metabolites for all samples. Disease status is indicated by the colour at the left side of each row. Panel (**c**) describes the taxonomic distribution at the phylum level of groups of species before and after community reduction, according to the disease status. Selection of communities was performed with the objective of making producible by the reduced communities the set of metabolites in the cooperation potential. init: initial composition of communities, KS: key species, AS: alternative symbionts, ES: essential symbionts. T1D: Type-1 Diabetes, T2D: Type-2 Diabetes.

