## [Decision Letter]

**Acceptance summary:**

Knowledge of the functions and interactions among members of a complex microbial community is crucial to understanding their roles and ecological relevance. This work presents a flexible workflow, M2M, tailored to the metabolic analysis of microbiomes from metagenomics data. It integrates several tools that allow metabolic modelling of large-scale communities, inferring metabolic complementarity and identification of species key to a given community.

**Decision letter after peer review:**

Thank you for submitting your article "Metage2Metabo: microbiota-scale metabolic complementarity for the identication of keystone species" for consideration by *eLife*. Your article has been reviewed by three peer reviewers, and the evaluation has been overseen by a Reviewing Editor and Gisela Storz as the Senior Editor. The following individuals involved in review of your submission have agreed to reveal their identity: Daniel Machado (Reviewer #1); Oliver Ebenhoh (Reviewer #2).

The reviewers have discussed the reviews with one another and the Reviewing Editor has drafted this decision to help you prepare a revised submission.

Summary:

This study presents a pipeline for large-scale metabolic analysis of genomes and metagenomic data. When evaluated on gut microbiome genomes from cultured species and metagenome-assembled genomes, this pipeline was able to reconstruct metabolic networks, identify potential metabolites, and provide information on minimal communities for a given target production and keystone species, defined here as species that are essential for a community to perform a certain metabolic function. The manuscript clearly formulates limitations, explains the software and functionality, and provides source code in a well-structured and clear git repository.

Essential revisions:

The paper could benefit from revisions to improve clarity, and to showcase the tool's versatility and performance. While aspects of the pipeline are novel, such as relying on topological methods and the use of Answer Set Programming to solve the problems of finding minimal sets of reactions or minimal sets of organisms, the authors need to revise some of the novelty claims in light of previous work and existing tools.

Please address the following concerns regarding the novelty of the pipeline and its automation, as well as comparison with existing tools and annotation:

1) There are already some tools that can automate GSMN reconstruction from genomes and MAGs (for example, ModelSEED and CarveMe, this one mentioned), calculate cooperation from GSMNs (SMETANA; Zelezniak et al., 2015), and workflows that automate some tasks such as metaBAGpipes (https://github.com/franciscozorrilla/metaBAGpipes), which assembles MAGs from raw metagenomics data, CarveMe, MEMOTE (model quality control) and SMETANA) and in KBase it is possible to assemble MAGs, run ModelSEED, and merge the single species models into community models for further analysis. These options go from raw metagenomic data (which the m2m software apparently doesn't) to community GSMN analysis and simulation.

2) It seems incorrect to state that their network-expansion algorithm can scale to large communities, unlike other simulation methods based on flux balance analysis. There are LP and MILP implementations of FBA community simulation methods. The LP methods scale linearly with the number of species (Popp and Centler, 2020 recently simulated a 773-species community with FBA). The MILP-based methods are worst-case exponential, but commercial solvers like CPLEX and Gurobi implement very efficient heuristics that allow for fast simulation. To make this claim, the authors should clearly state what is the computational cost (using Big O notation) of their method and show that it is lower than FBA based methods.

3) Different databases are functionally annotated by different tools (Prokka and EGGNOG). Annotation method for the third dataset is unclear. Different methods are employed on different datasets which in turn might hurt the analysis due to lack of standardization of the inputs.

4) Please clarify what is meant when stating that their network expansion algorithm is more robust "in the face of missing reactions". If this means that the algorithm doesn't fail to compute, then it might lead to results that may represent an incorrect metabolic landscape. As such, robustness may not necessarily be correct (or desirable). It could also be that the metabolic end-products obtained with FBA (by performing flux variability analysis of the exchange reactions) would in the end be the same, since in both cases they have to be topologically reachable regardless of the stoichiometry being accounted for or not.

5) While the authors mention reasons for robustness of their algorithms, they do not test these possibilities. Perhaps these could be addressed given that they have all the data required to answer the question.

Please take into account the following points in order to improve the manuscript.

6) The Materials and methods section could benefit from a major revision as it is going back and forth from describing the datasets and the pipeline (the focus is unclear). It'll be useful to have more details on critical steps in the pipeline such as the metabolic objective and community reduction steps and keystone species discovery.

7) The average number of metabolites per model is higher than the average number of reactions (1366 vs 1144 for the gut dataset). GSMNs usually have a lot more reactions than metabolites, hence their underdetermined stoichiometric matrices, and large number of degrees of freedom. Also, the average number of reachable metabolites is only 286, i.e. only about 20% of the metabolic network is reached. How is this possible, and how can one trust such models?

8) There are multiple instances in the Results section where the authors present the p-value for a statistical test without presenting also the effect size and/or test statistic. Knowing the statistical significance is not helpful without knowing the effect size. For instance: "The community diversity varied between disease statuses, with a significantly higher number of MGS observed in T1D individuals forming the initial communities (anova p < 0.01, Tukey HSD test p < 0.01 vs control)."

9) The authors mention that "A classification experiment on the composition of the community scope can, to some extent, (AUC = 0.73 +/- 0.15) decipher between healthy or diabetes statuses." But is this better than a classification based, for instance, on OTU analysis, or functional meta-genome analysis? Although the results are interesting, in the end it is hard to convince the reader that using GSMN reconstruction provides an advantage compared to using the metagenomics data directly.

10) In general, results are not compared to the state of the art and the Results section should contain more specific examples of metabolites/pathways of interest, bacterial species and their known or novel interactions. Additionally, it is mentioned that the datasets are similar – it'll be useful to have a section summarizing the results from all analyses.

11) The utilization of keystone species in this work is not entirely correct. Here the authors use keystone species to mention species that are always present in the set of minimal communities enumerated to produce a given set of metabolites. The definition of keystone species in a community are those whose removal would cause the collapse of the community. Since the simulation method used by the authors doesn't allow to test for community stability, the application of this term does not seem appropriate.

12) Keystone species are also described as the output of the tool and could use a more detailed report and examples in the Results section. These are very interesting and currently get lost between the lines.

13) There are several aspects of the figures that can be improved.

– Figure 1 is confusing could use some reorganization, so the pipeline steps are clear, consider adding numbers to the different steps.

– Figure 2 is very hard to digest. I have difficulties understanding what the figure actually tells me. What is the meaning of the white fields, are the sub-figures connected despite having a different x-axis, and what is the overall message?

– The power graphs are interesting, but it is unclear if they were generated by the tool since this is not clear in the manuscript. In addition, the usefulness of the power graphs in Figure 3 is not fully evident. What do we learn from them and what are the large number of circles? If three subgroups are connected, why are two of them encircled in an extra circle?

– Figure 4 presents an analysis that is downstream of the presented software paper since it illustrates how the output of the software can be further analyzed. In order to appear in the main text of the manuscript, it needs to be better explained since it is hard to understand with the information given. For example, what do the Receiver Operator Curves (ROC) actually represent? More background information is required.

14) The authors should define the essential and non-essential symbionts and add more context on their known interactions in the Introduction and Results sections.

15) The comparison to other platforms such as Kbase and other genome scale models should be discussed in more detail in the Introduction and Discussion sections. It is unclear how this tool can make use of available good quality curated reconstructions as input.

---

## [Author Response]

Essential revisions:The paper could benefit from revisions to improve clarity, and to showcase the tool's versatility and performance. While aspects of the pipeline are novel, such as relying on topological methods and the use of Answer Set Programming to solve the problems of finding minimal sets of reactions or minimal sets of organisms, the authors need to revise some of the novelty claims in light of previous work and existing tools.

In this manuscript we present Metage2Metabo, a flexible workflow dedicated to the metabolic screening of large communities. This workflow assembles several steps of analysis starting from the automation of metabolic network reconstruction with Pathway Tools. This step of command-line automation of Pathway Tools is a novelty of the paper, we developed a Python package ensuring a transparent reconstruction for a set of annotated genomes. The subsequent analyses of the workflow rely on the network expansion algorithm that was previously published (Ebenhöh et al., 2004). Modelling of network expansion in Answer Set Programming has been previously applied to gap-filling (Schaub and Thiele, 2009, Prigent et al., 2017). Here we designed tools to directly apply network expansion to the study of individual GSMNs in order to retrieve activated reactions and producible metabolites. In addition, an Answer Set Programming algorithm relying on network expansion is used for community reduction. This particular algorithm was already published in 2018 (Frioux et al., Bioinformatics) as a Python package (MiSCoTo). Data management, encapsulation and outputs of MiSCoTo were extensively enhanced in the context of M2M development. The core algorithm remained, but we added logic programming enhancements notably for the identification of target metabolite producers in initial or selected communities. In addition, the use of the algorithm in M2M is facilitated as it occurs in the general context of the pipeline and results are directly analysed in the light of the initial microbiota. The combination of all these individual tools into a complete and automatic, yet flexible, pipeline is a novelty of the manuscript: performing individual then collective analysis of the metabolic potential, identifying the newly producible metabolites as a cooperation potential, defining key species and essential/alternative symbionts by analysing the search space of minimal communities, validating the usage to metagenome-derived datasets.

We used the comments and advice of the reviewers to enhance the manuscript and clarify the novelty it brings. We have notably reorganised the Materials and methods (now at the end of the manuscript) with respect to the Result section. We mathematically formalised the concepts in the methods, and clarified the new concepts brought by the manuscript: key species, essential and alternative symbionts, cooperation potential. We reorganised the results around the main object of the manuscript which is the pipeline: how does every experiment gives information on the usability and applicability of the pipeline. Finally, we rewrote parts of the Discussion to underline important comments raised by the reviewers. Altogether, we believe the manuscript gained clarity after this step of review and we thank the reviewers for leading us in this direction.

Please address the following concerns regarding the novelty of the pipeline and its automation, as well as comparison with existing tools and annotation:1) There are already some tools that can automate GSMN reconstruction from genomes and MAGs (for example, ModelSEED and CarveMe, this one mentioned), calculate cooperation from GSMNs (SMETANA; Zelezniak et al., 2015), and workflows that automate some tasks such as metaBAGpipes (https://github.com/franciscozorrilla/metaBAGpipes), which assembles MAGs from raw metagenomics data, CarveMe, MEMOTE (model quality control) and SMETANA) and in KBase it is possible to assemble MAGs, run ModelSEED, and merge the single species models into community models for further analysis. These options go from raw metagenomic data (which the m2m software apparently doesn't) to community GSMN analysis and simulation.

We agree with the observations of the reviewers and clarified the existing tools in the manuscript. We want to emphasize that we designed M2M to be flexible and suitable to metabolic models of any database, reconstructed with any tool. A module of M2M focuses on the automation of Pathway Tools to fill a gap in the usage of the software at a large scale. Pathway Tools is a widely used tool for GSMN reconstruction, we aimed at facilitating its usage from command-line in order to ease its scalability. The other tools, as stated by reviewers can already be used with these objectives, and possibly extend the usage to raw metagenomic data.

The references for KBase, SMETANA, ModelSeed were all already provided in the Introduction, although the names of the tools were not explicitely cited. This is the case now.

“There now exist a variety of GSMN reconstruction implementations: all-in-one platforms such as Pathway Tools (Karp et al., 2016), CarveMe (Machado et al., 2018a) or KBase that provides narratives from metagenomic datasets analysis up to GSMN reconstruction with ModelSeed (Henry et al., 2010)”

“SMETANA (Zelezniak et al., 2015) estimates the cooperation potential and simulates fluxes exchanges within communities. MiSCoTo (Frioux et al., 2018b) computes the metabolic potential of interacting species and performs community reduction. NetCooperate (Levy et al., 2015) predicts the metabolic complementarity between species.”

We emphasised in the Discussion on the possible use of metabolic models from other databases.

“GSMNs in SBML format obtained from other platforms such as Kbase (Arkin et al., 2018), ModelSEED (Henry et al., 2010) or CarveMe (Machado et al., 2018a), can also be used as inputs to M2M for all metabolic analyses. For instance, we used highly curated models from AGORA in the application of M2M to metagenomic datasets. The above reconstruction platforms already implement solution to facilitate the treatment of large genomic collections. There is no universal implementation for GSMN reconstruction (Mendoza et al., 2019); depending on their needs (local run, external platform, curated or non-curated GSMNs…), users can choose either method and connect it to M2M.”

2) It seems incorrect to state that their network-expansion algorithm can scale to large communities, unlike other simulation methods based on flux balance analysis. There are LP and MILP implementations of FBA community simulation methods. The LP methods scale linearly with the number of species (Popp and Centler, 2020 recently simulated a 773-species community with FBA). The MILP-based methods are worst-case exponential, but commercial solvers like CPLEX and Gurobi implement very efficient heuristics that allow for fast simulation. To make this claim, the authors should clearly state what is the computational cost (using Big O notation) of their method and show that it is lower than FBA based methods.

M2M falls in the category of optimization problems for which worst-case asymptotic complexity is not as informative as practical performance using heuristic methods. The same is true of mixed-integer linear programming methods.

As a strategy layer over MiSCoTo (Frioux et al., 2018), metage2metabo relies on the Clasp solver (Gebser et al., 2012) for efficient resolution of the underlying grounded ASP instances. Although this type of decision problem is NP-hard (Julien-Laferriere et al., 2016), as with many real-world optimisation problems worst-case asymptomatic complexity is less informative for applications than practical performance using heuristic methods. The Clasp solver implements a robust collection of heuristics (Gebser et al., 2007, Andres et al., 2012) for core-guided weighted MaxSAT (Manquinho et al., 2009, Morgado et al., 2012) that provide rapid set-based solutions to combinatorial optimization problems, much in the same way that heuristic polytope solvers like CPlex [www.ibm.com/analytics/cplex-optimizer] provide rapid numerical solutions to mixed integer programming optimization problems. The kinds of ASP instances constructed by MiSCoTo for metage2metabo are solved in a matter of minutes. In addition, the computation of key species uses projection modes (brave and cautious reasoning) that do not necessitate the enumeration of solutions. These reasoning modes are used in addition to the solving heuristics that enable the solving of the optimisation problem in the first place.

We thank the reviewers for pointing out the work of Popp and Centler. Let us emphasize that we do not think M2M has the exact same usage as such methods. M2M is suitable for screening, applicable in the absence of biomass reaction, and easily ran, with computation lengths of minutes without the reconstruction part, with various target metabolites. Simulations as in the above article take days and are, to our opinion, less flexible in the set-up than M2M, which makes us suggest to use them in second intention, after screening of the community metabolic complementarity with M2M, and possibly pre-isolation of species of interest. Recent works also emphasise on the difficulty of using simulations taking many species into account. For example, Basile et al. (2020) instead performed pairwise simulations.

Finally, in our case, the optimisation occurs in the community reduction step. Finding what is producible in each community does not require optimisation, it is governed by logic rules in Answer Set Programming. Julien-Laferrière and co-workers designed an algorithm solving the Directed Steiner Hypertree problem for the design of minimal synthetic consortia and studied its complexity. In practice however, it was the description of the problem in ASP that led to an efficient solving.

“As a strategy layer over MiSCoTo, M2M relies on the Clasp solver (Gebser et al., 2012) for efficient resolution of the underlying grounded ASP instances. Although this type of decision problem is NP-hard (Julien-Laferriere et al., 2016), as with many real-world optimisation problems worst-case asymptomatic complexity is less informative for applications than practical performance using heuristic methods. The Clasp solver implements a robust collection of heuristics (Gebser et al., 2007, Andres et al., 2012) for core-guided weighted MaxSAT (Manquinho et al., 2009, Morgado et al., 2012) that provide rapid set-based solutions to combinatorial optimisation problems, much in the same way that heuristic solvers like CPlex provide rapid numerical solutions to mixed integer programming optimisation problems. The kinds of ASP instances constructed by MiSCoTo for M2M are solved in a matter of minutes for the identification of key species and essential/alternative symbionts. Indeed the space of solutions is efficiently sampled using adequate projection modes in ASP, which enables the computation of these groups of species without the need for a full enumeration by taking advantage of the underlying ASP solver and associated projection modes.”

In addition, we clarified our statement in Discussion:

“Most metabolic modelling methods rely on flux analyses (Orth et al., 2010) solved with linear programming, which may turn out to be challenging to implement for simulations of large communities (Basile et al., 2020), although recent efforts in that direction are encouraging (Popp et al., 2020). M2M uses the network expansion algorithm and solves combinatorial optimisation problems with Answer Set Programming, thereby ensuring fast simulations and community predictions, suitable when performing systematic screening and multiple experiments.”

3) Different databases are functionally annotated by different tools (Prokka and EGGNOG). Annotation method for the third dataset is unclear. Different methods are employed on different datasets which in turn might hurt the analysis due to lack of standardization of the inputs.

M2M expects as input annotated genomes in GenBank formats (with m2m workflow) or SBML format (with m2m metacom). We have applied the tool to four datasets, three with GenBank as inputs (gut, rumen and diabete datasets) and one with SBML as input (AGORA datasets).

The GenBanks were obtained using several annotation pipelines depending on the input data. All the annotation pipelines we used were previously published and are extensively used by the community. The genomic data differed for each dataset: one consisted in already annotated genomes (gut reference genomes dataset), one was a collection of contigs binned into MAGs (rumen dataset) and the last one consisted in lists of genes for metagenomic species (diabetes dataset).

The gut reference genomes was already annotated by the authors of the work. So we directly reused this annotation by feeding the annotated GenBank files extracted from NCBI to M2M.

Regarding the rumen dataset, the initial dataset consisted in a set of contigs for each MAG. Among the pipelines suitable for contig annotation, Prokka is widely used. We annotated the MAGs using Prokka and directly used the GenBank created by the tool as an input to M2M.

Finally in the diabetes dataset, the data consisted in core genes of MGS on one hand, and for species that could be mapped to the Progenomes database, in the representative genome of each, for which we downloaded the genes and proteins. We used Eggnog-mapper to functionally annotate the genes. Then with these lists of genes and their annotations we used the package emapper_to_gbk to produce GenBank file compatible with M2M.

“Users can use the annotation pipeline of their choice prior running M2M.”

“The genomes were already annotated and could therefore directly enter M2M pipeline.”

4) Please clarify what is meant when stating that their network expansion algorithm is more robust "in the face of missing reactions". If this means that the algorithm doesn't fail to compute, then it might lead to results that may represent an incorrect metabolic landscape. As such, robustness may not necessarily be correct (or desirable). It could also be that the metabolic end-products obtained with FBA (by performing flux variability analysis of the exchange reactions) would in the end be the same, since in both cases they have to be topologically reachable regardless of the stoichiometry being accounted for or not.

Our objective with this sentence was to highlight previous results of network expansion analyses that demonstrated the stability of the scope even in the case of missing reactions in the metabolic network. This seemed adequate in the metagenomic context as genomes used for GSMN reconstruction are associated to species or strains that are poorly studied and can therefore be incomplete or face incomplete annotation. In addition, the metabolic network draft automatically constructed can have gaps that one does not want to automatically fill as it could lead to adding possibly false positive reactions that would hide metabolic complementarity with other species.

We rewrote the sentence in the Introduction.

“The robustness of the algorithm was demonstrated by the stability of the set of reachable metabolites despite missing reactions (Handorf et al., 2005, Kruse et al., 2008)”

5) While the authors mention reasons for robustness of their algorithms, they do not test these possibilities. Perhaps these could be addressed given that they have all the data required to answer the question.

The second hypothesis described in the text, which is the robustness of the algorithm to missing reactions, has already been validated and tested in the context of robustness (Handorf et al., 2005, Kruse et al., 2008). In that sense, it seems out of the scope of the paper to make an additional validation of the algorithm.

Regarding the other possibilities we proposed, we calculated the effect of genome degradation on the resulting GSMNs. The results are in Supplementary file 1—tables 16 and 21-23. Interestingly, the number of genes included in metabolic networks follows a decrease that is similar to the decrease of genes in the original genomes. So our first hypothesis, degraded genes are not related to metabolism, is refuted. The average number of reactions and metabolites in GSMNs decreases in degraded metabolic networks but the decrease is slower than the gene loss in associated MAGs. For instance in the experiment where 70% of genomes were 10 percent degraded, the average loss of genes in the dataset is of 6.94%, similar to the gene loss in GSMNs (7.02%). The average reaction loss in associated GSMNs is only of 3.92%. We also observe that the percentage of reactions associated to a gene is remarkably stable in all experiments. This suggests that the loss of reactions mainly occurs among reactions that are not associated to genes. It is also possible that the loss of genes in GSMNs is due to the redundancy loss: some reactions associated to several genes before degradation lose some of these gene associations after degradation.

We also added a few sentences in the main text, with reference to the Appendix.

“Additional analyses (Appendix 1) enable the refutation of the first hypothesis as the average gene loss in metabolic networks is similar to the genomic loss. Yet, the percentage of reactions associated to genes is similar in every experiments, which goes in the direction of the redundancy loss hypothesis. Likewise, we observed that the loss in reactions for degraded genomes is lower than the loss of genes.”

Appendix 1: Table 1 + the following text:

“MAGs from the rumen dataset were degraded by randomly removing contigs. The following degradations were tested: removal of 2% of genes in all MAGs, removal of 5% of genes in 80% of MAGs, removal of 5% of genes in all MAGs, removal of 10% of genes in 70% of MAGs. Table 1 summarises the characteristics of the genomes and GSMNs for all experiments. The average gene loss in genomes is similar to the average gene loss in metabolic networks. However, the average loss of metabolites and reactions is lower than the genetic loss: it increases more slowly than the loss of genes. For instance, the 2-percent degradation of MAGs leads to a nearly 2 percent decrease in reaction numbers in GSMNs. However, the 10-percent degradation in 70% of genomes (average gene loss of 7% in the initial community) only leads to a 5% decrease in reaction numbers. One notable observation is the stability in the percentage of reactions associated to genes, suggesting that the loss of reactions in degraded genomes mainly occurs among reactions that are not associated to genes. It is also possible that the loss of genes in GSMNs is due to the redundancy loss: some reactions associated to several genes before degradation lose some of these gene associations after degradation. Data for each genome and GSMN is available in Supplementary file 1—tables 16, 21-24.”

Please take into account the following points in order to improve the manuscript.6) The methods section could benefit from a major revision as it is going back and forth from describing the datasets and the pipeline (the focus is unclear). It'll be useful to have more details on critical steps in the pipeline such as the metabolic objective and community reduction steps and keystone species discovery.

We thank the reviewers for the suggestion. We accordingly reorganised the sections and subsections. The Materials and methods are now at the end of the manuscript. Key concepts brought by the paper are moved in the first part of the Results section. We clarified the organisation of the Materials and methods. There is now one main section dedicated to the implementation of the pipeline, with subsections addressing its main steps, and mathematical formalisms to describe what is done by each step. The second part of the Materials and methods now describes the details regarding the applications of M2M to various datasets. One subsection describes the two collections of genomes and MAGs, including protocols for the robustness analysis, and a second subsection on the de novo reconstruction of MAGs and subsequent M2M analysis on individual samples.

7) The average number of metabolites per model is higher than the average number of reactions (1366 vs 1144 for the gut dataset). GSMNs usually have a lot more reactions than metabolites, hence their underdetermined stoichiometric matrices, and large number of degrees of freedom. Also, the average number of reachable metabolites is only 286, i.e. only about 20% of the metabolic network is reached. How is this possible, and how can one trust such models?

In order to build the SBML for the metabolic networks associated to the collections of genomes, the Padmet dependency of M2M (Aite et al., 2018) retrieves the list of metabolic reactions predicted by Pathway Tools and available in the reactions.dat file of the PGDBs. We built the list of metabolites by retrieving the reactants and products of all the reactions. That list of metabolites is consistently larger than the number of reactions for all PGDBs built by Pathway Tools. The reason for this observation is that metabolites contains both regular species, and classes of metabolites (for example “a-fatty-acid”). This therefore artificially increases the ratio of the number of metabolites to the number of reactions. We did not want to systematically “instantiate” each reaction involving a class of metabolites with all the instances of that class to prevent adding reactions that would not have genetic support.

The reachable metabolites as stated in the manuscript do not indicate that no other metabolite of the model could be producible. It is directly bound to the seeds (nutrients) that are provided to the algorithm. We chose small sets of nutrients for the gut reference genomes and rumen MAGs analyses. In addition to the small sets of seeds, it has to be noted that GSMN are considered individually for the computation of these metrics. In an ecosystem context, the associated species would instead benefit from the changes in the medium brought by the other species, widening the set of available nutrients. Additionally, the metabolic models are not gap-filled nor curated thereby not hiding the putative auxotrophies of the species that are not covered by our minimal medium (Bernstein et al., 2018).

8) There are multiple instances in the Results section where the authors present the p-value for a statistical test without presenting also the effect size and/or test statistic. Knowing the statistical significance is not helpful without knowing the effect size. For instance: "The community diversity varied between disease statuses, with a significantly higher number of MGS observed in T1D individuals forming the initial communities (anova p < 0.01, Tukey HSD test p < 0.01 vs control)."

We apologise for the missing information. We added the test statistic value and the Eta squared to all of our tests. Additionally, the tests and their results are summarised in Supplementary file 1—table 17.

9) The authors mention that "A classification experiment on the composition of the community scope can, to some extent, (AUC = 0.73 +/- 0.15) decipher between healthy or diabetes statuses." But is this better than a classification based, for instance, on OTU analysis, or functional meta-genome analysis? Although the results are interesting, in the end it is hard to convince the reader that using GSMN reconstruction provides an advantage compared to using the metagenomics data directly.

We did not want this analysis to illustrate an advantage of GSMN reconstruction. We rather wanted to illustrate that despite metabolic redundancy in the gut microbiota and the automatic reconstruction of metabolic networks, qualitative differences are still noticeable at the level of metabolism between healthy and diabetic individuals: it is possible to distinguish to some extent the set of metabolites predicted to be producible by the microorganisms found in their faeces.

“Although metagenomic data would more precisely perform such a separation, it is informative to observe that despite metabolic redundancy in the gut microbiota, there are differences at the metabolic modelling level. Qualitative differences are noticeable between healthy and diabetic individuals: it is possible to distinguish them to some extent using the set of metabolites predicted to be producible by the microorganisms found in their faeces.”

10) In general, results are not compared to the state of the art and the Results section should contain more specific examples of metabolites/pathways of interest, bacterial species and their known or novel interactions. Additionally, it is mentioned that the datasets are similar – it'll be useful to have a section summarizing the results from all analyses.

We performed a literature analysis of species identified as essential symbionts in the gut collection of reference genomes and further studied with the power graph analysis. We retrieved some known features of species, consistent with our predictions, notably the ability of the *Burkholderiales* order to degrade aromatic compounds, or the variety of carbohydrates that *Paenibacillus polymyxa* could degrade.

We added the following paragraph to our results:

“We further investigated the essential symbionts associated to carbohydrate-derived metabolites in our study: *Paenibacillus polymyxa*, *Lactobacilluslactis*, *Bacilluslicheniformis*, *Lactobacillusplantarum*, and *Dorea longicatena*. […] Finally, the only essential symbiont predicted for the coA-related metabolites is Fusobacterium varium, a butyrate producer known for its ability to ferment both sugars and amino acids (Potrykus et al.,2008).”

In addition, we added the third dataset (diabetes) results to the Table 2 so that the three datasets can be compared together.

We understand the comment of the reviewers regarding comparison to state of the art methods. The underlying algorithms of M2M have already been validated and compared: the network expansion algorithm, but also the community selection algorithm (Frioux et al. Bioinformatics 2018) so we believe their comparison is out of the scope of the paper. We identified MICOM as a suitable pipeline for comparison to M2M in this manuscript and illustrated that despite the limited comparability of both tools, there are similarities between their predictions.

11) The utilization of keystone species in this work is not entirely correct. Here the authors use keystone species to mention species that are always present in the set of minimal communities enumerated to produce a given set of metabolites. The definition of keystone species in a community are those whose removal would cause the collapse of the community. Since the simulation method used by the authors doesn't allow to test for community stability, the application of this term does not seem appropriate.

We understand the concern raised by the reviewers. The ecological concept of keystone species used in the literature would be closer to our concept of essential symbiont, although the latter one is defined in a context of *minimal* communities. In order to remove this confusion, we chose “key species” to describe what we previously described as keystone species. We believe that while being close to the old name which will retain consistency for our users, this change removes the possible misunderstanding between the ecological concept and our output.

We therefore removed “keystone species” from the title, the figures and the manuscript and replaced it with “key species”.

In addition, we discussed these concepts:

“The identification of cornerstone taxa in microbiota is a challenge with many applications, for instance restoring balance in dysbiotic environments. Keystone species in particular are thoroughly looked for as they are key drivers of communities with respect to functions of interest (Banerjee et al., 2018). There is a variety of techniques to identify them (Carlstrom et al., 2019, Floch et al., 2020), and computational biology has a major role in it (Fisher et al., 2014, Berry et al., 2014). The identification of alternative and essential symbionts by M2M is an additional solution to help identify these critical species. In particular, essential symbionts are close to the concept of keystone species as they are predicted to have a role in every minimal community associated to a function. Additionally, alternative species and the study of their combinations in minimal communities, for example with power graphs, are also informative as they reveal equivalence groups among species.”

12) Keystone species are also described as the output of the tool and could use a more detailed report and examples in the Results section. These are very interesting and currently get lost between the lines.

The novel concepts brought by M2M on essential and alternative symbionts were described in Materials and methods but not in results in the first submission of the manuscript. We reorganised these sections and notably added a first result section that describes the pipeline *per se* as it is the main novelty brought by the paper. We describe the steps of the pipelines, its flexibility and focus on the definition of key species. The latter is also illustrated in a subfigure of Figure 1 to ease the understanding of the concepts on a small example. We also added in the Materials and methods section mathematical descriptions of the concepts used in the pipelines.

“A main characteristic of M2M is to provide at the end of the pipeline a set of key species associated to a metabolic function together with one minimal community predicted to satisfy this function. We define as key species organisms whose GSMNs s are selected in at least one of the minimal communities predicted to fulfil the metabolic objective. Among key species, we distinguish those that occur in every minimal community, suggesting that they possess key functions associated to the objective, from those that occur only in some communities. We call the former essential symbionts, and the latter alternative symbionts. These terms were inspired by the terminology used in flux variability analysis (Orth et al., 2010) for the description of reactions in all optimal flux distributions. If interested, one can compute the enumeration of all minimal communities with m2m-analysis, which will provide the total number of minimal communities as well as the composition of each. Figure 1B. illustrates these concepts. The initial community is formed of eight species. There are four minimal communities satisfying the metabolic objective. Each includes three species, and in particular the yellow one is systematically a member. Therefore the yellow species is an essential symbiont whereas the four other species involved in minimal communities constitute the set of alternative symbiont. As key species represent the diversity associated to all minimal communities, it is likely that their number is greater than the size of a minimal community, as this is the case in Figure 1B.”

13) There are several aspects of the figures that can be improved.– Figure 1 is confusing could use some reorganization, so the pipeline steps are clear, consider adding numbers to the different steps.

Thank you for the suggestion. Accordingly, we reorganised Figure 1 with numbered steps and boxes to distinguish them. In addition, we illustrated the concept of key species, essential symbionts and alternative symbionts in Figure 1B and in the beginning of the Results section.

“Legend of Figure 1: a. Main steps of the M2M pipeline and associated tools. The software's main pipeline (m2m workflow) takes as inputs a collection of annotated genomes that can be reference genomes or metagenomics-assembled genomes. The first step of M2M consists in reconstructing metabolic networks with Pathway Tools (step 0). This first step can be bypassed and GSMN s can be directly loaded in M2M. The resulting metabolic networks are analysed to identify individual (step 1) and collective (step 2) metabolic capabilities. The added-value of cooperation is calculated (step 3) and used as a metabolic objective to compute a minimal community and key species (step 4). Optionally, one can customise the metabolic targets for community reduction. The pipeline without GSMN reconstruction can be called with m2m metacom, and each step can also be called independently (m2m iscope, m2m cscope, m2m added value, m2m mincom). b. Description of key species. Community reduction performed at step 4 can lead to multiple equivalent communities. M2M provides one minimal community and efficiently computes the full set of species that occur in all minimal communities, without the need for a full enumeration, thanks to solving heuristics. It is possible to distinguish the species occurring in every minimal community (essential symbionts), from those occurring in some (alternative symbionts). Altogether, these two groups form the key species.”

– Figure 2 is very hard to digest. I have difficulties understanding what the figure actually tells me. What is the meaning of the white fields, are the sub-figures connected despite having a different x-axis, and what is the overall message?

The representations in Figure 2 are alternatives to Venn diagrams. The white fields indicate missing elements in the corresponding dataset. For example in a), there is a set of 37 metabolites (on the right) that the original dataset can produce, but not the other datasets. On the left of subfigure c), there is a set of 11 species that are key species in all datasets except in the one where all genomes were degraded at 5%. We added examples to ease the reading in subfigure a). All subfigures illustrate some information that M2M provided when run on each of the five datasets. We added a sentence to clarify the independence between all subfigures. We removed the “Number of” in the legend titles of all subfigures. The overall message from this figure is the relative stability of the information computed by M2M to small degradations of genomes.

“Legend of Figure 2: Subfigures A to E each represent one piece of information computed by M2M and compared between the five experiments. […] Vertical overlaps between sets represent intersections (e.g groups of metabolites retrieved in several datasets) whose size is indicated on the X axis. For example, there is a set of 37 metabolites that are producible in the original dataset only, and a set of 5 metabolites predicted as producible in all datasets but the one where 70% of genomes were 10%-degraded. A full superimposition of all the coloured bars would indicate a complete stability of the community scope between datasets.”

“Altogether, Figure 2 illustrates relative stability of the information computed by M2M to missing genes. The criteria typically used for MAG quality (>80% completeness, <10% degradation) are likely sufficient to get a coarse-grained, yet valuable first picture of the metabolism.”

– The power graphs are interesting, but it is unclear if they were generated by the tool since this is not clear in the manuscript. In addition, the usefulness of the power graphs in Figure 3 is not fully evident. What do we learn from them and what are the large number of circles? If three subgroups are connected, why are two of them encircled in an extra circle?

We implemented the generation of power graphs in *m2m-analysis*. We added this information in the legend of Figure 3. The colouring of the power graph in Figure 3 was nonetheless done manually. The power graph analysis is informative as it pinpoints groups of equivalent species among the key species. Visualising the association between members of thousands of equivalent minimal communities is a challenge. Here we compressed graph describing these associations. Specific patterns from the original graph are therefore illustrated as links between power nodes (the circles) and we retrieve in the power nodes key species that play the same role in all minimal communities (and therefore are interchangeable with respect to the metabolic objective). Some circles can be included into others to limit the number of edges to be represented. Indeed, there is not a unique representation of a power graph, the algorithm we use provides one solution. The number of power edges is minimal, which leads to nesting of (power) nodes. For example, in power graph a), power node 1 is linked to power nodes 2, 3, 4 and 5. Including the latter four in a larger power node makes it possible to represent the associations with a single edge.

We added details in the legend of Figure 3. We also explicated that power graphs can be created by m2m-analysis in the Materials and methods.

– Figure 4 presents an analysis that is downstream of the presented software paper since it illustrates how the output of the software can be further analyzed. In order to appear in the main text of the manuscript, it needs to be better explained since it is hard to understand with the information given. For example, what do the Receiver Operator Curves (ROC) actually represent? More background information is required.

We added some context on the analysis in the beginning of the legend: “M2M was run on collections of GSMNs associated to MAGs identified in metagenomic samples from a cohort of healthy and diabetic individuals.”. The experiment that led to the ROC curve aimed at identifying whether differences occur in the contents of producible metabolites between healthy and diseased individuals. Indeed, it is not straightforward whether the set of producible metabolites predicted for the intestinal community would qualitatively differ between healthy and diseased individual, notably because the metabolism of the gut microbiota is redundant, and because differences are likely to be quantitative. Here, the classification experiments that there are qualitative differences, that can to some extent, distinguish the two categories. “Figure d is the receiver operating curve (ROC) of a SVM classification experiment aiming at predicting the disease status for the MHD cohort (control n=49 or diabetes n=66) based on the community scope composition.”

14) The authors should define the essential and non-essential symbionts and add more context on their known interactions in the introduction and Results sections.

The novel concepts brought by M2M on essential and alternative symbionts were described in Materials and methods but not in results in the first submission of the manuscript. We reorganised these sections in order to define key concepts that are novel in the results.

We added a subfigure to Figure 1 to illustrate the concept of key species, essential and alternative symbionts.

“A main characteristic of M2M is to provide at the end of the pipeline a set of key species associated to a metabolic function together with one minimal community predicted to satisfy this function. We define as key species organisms whose GSMNs s are selected in at least one of the minimal communities predicted to fulfil the metabolic objective. Among key species, we distinguish those that occur in every minimal community, suggesting that they possess key functions associated to the objective, from those that occur only in some communities. We call the former essential symbionts, and the latter alternative symbionts. These terms were inspired by the terminology used in flux variability analysis (Orth et al., 2010) for the description of reactions in all optimal flux distributions. If interested, one can compute the enumeration of all minimal communities with m2m-analysis, which will provide the total number of minimal communities as well as the composition of each. Figure 1B. illustrates these concepts. The initial community is formed of eight species. There are four minimal communities satisfying the metabolic objective. Each includes three species, and in particular the yellow one is systematically a member. Therefore the yellow species is an essential symbiont whereas the four other species involved in minimal communities constitute the set of alternative symbiont. As key species represent the diversity associated to all minimal communities, it is likely that their number is greater than the size of a minimal community, as this is the case in Figure 1B.”

15) The comparison to other platforms such as Kbase and other genome scale models should be discussed in more detail in the introduction and Discussion sections. It is unclear how this tool can make use of available good quality curated reconstructions as input.

M2M proposes a solution based on the MetaCyc database and its associated software Pathway Tools for GSMN reconstruction. We developed this solution to facilitate the automatic use of Pathway Tools on a large number of genomes. However, we are aware that several software solutions coexist for this objective. This is why one can provide to M2M SBML files of metabolic networks that were previously reconstructed. This is what we did when we tested the AGORA metabolic models on the diabetes dataset. The analyses of M2M without reconstruction can be done with the m2m metacom command.

We clarified the possibility to use external data with M2M in the legend of Figure 1.

Legend of Figure 1: “The first step of M2M consists in reconstructing metabolic networks with Pathway Tools (step 0). This first step can be bypassed and GSMNs can be directly loaded in M2M.”

“GSMNs obtained from other platforms such as Kbase (Arkin et al., 2018), ModelSEED (Henry et al., 2010) or CarveMe (Machado et al., 2018a), can also be used as inputs to M2M for all metabolic analyses. For instance, we used highly curated models from AGORA in the application of M2M to metagenomic datasets. The above reconstruction platforms already implement solution to facilitate the treatment of large genomic collections. There is no universal implementation for GSMN reconstruction (Mendoza et al., 2019); depending on their needs (local run, external platform, curated or non-curated GSMNs…), users can choose either method and connect it to M2M.”